# Do LLMs "know" internally when they follow instructions?

**Juyeon Heo**[1,*]  **Christina Heinze-Deml**[2]  **Oussama Elachqar**[2]  **Kwan Ho Ryan Chan**[3,*]  **Shirley Ren**[2]
**Udhay Nallasamy**[2]  **Andy Miller**[2]  **Jaya Narain**[2]
[1]University of Cambridge   [2]Apple   [3]University of Pennsylvania
jh2324@cam.ac.uk  jnarain@apple.com

## ABSTRACT

Instruction-following is crucial for building AI agents with large language models (LLMs), as these models must adhere strictly to user-provided constraints and guidelines. However, LLMs often fail to follow even simple and clear instructions. To improve instruction-following behavior and prevent undesirable outputs, a deeper understanding of how LLMs' internal states relate to these outcomes is required. In this work, we investigate whether LLMs encode information in their representations that correlates with instruction-following success—a property we term "knowing internally". Our analysis identifies a direction in the input embedding space, termed the instruction-following dimension, that predicts whether a response will comply with a given instruction. We find that this dimension generalizes well across unseen tasks but not across unseen instruction types. We demonstrate that modifying representations along this dimension improves instruction-following success rates compared to random changes, without compromising response quality. Further investigation reveals that this dimension is more closely related to the phrasing of prompts rather than the inherent difficulty of the task or instructions. This work provides insight into the internal workings of LLMs' instruction-following, paving the way for reliable LLM agents.[1]

## 1  INTRODUCTION

Given the potential of large language models (LLMs), there has been significant interest in utilizing these models to build personal AI agents. For instance, one could imagine deploying an LLM as a personal healthcare assistant, such as a fitness or nutrition planner, or for psychological counseling (Li et al., 2024b; Wang et al., 2023; Tu et al., 2024). Compared to traditional machine learning-based AI agents, LLMs offer the advantage of being easily adaptable through prompting, allowing users to provide guidelines and personal information without the need to retrain model weights.

Instruction-following is critical in the development of personal AI agents with LLMs through prompts because these models must adhere to the constraints and guidelines to ensure safe and trustworthy interactions. For example, suppose an LLM is building a personal fitness plan for a user with knee problems. To avoid knee problems for the user, the LLM must follow the instruction of not recommending knee-intensive movements or any exercises that could lead to potential injury. Similarly, in a nutrition planner, the LLM should avoid generating harmful recommendations, such as suggesting inappropriate food for pregnant women or children with diabetes.

However, LLMs often fail to follow even unambiguous and simple instructions (Zhou et al., 2023; Qin et al., 2024; Xia et al., 2024; Kim et al., 2024; Yan et al., 2024) like including keywords or following formatting guidelines. GPT-4 achieves around an 80% success rate on IFEval (Zhou et al., 2023), an instruction-following benchmark dataset, while smaller models have success rates around 30% to 40%. This raises the question: why do LLMs fail to follow instructions, even when those instructions are clear and familiar?

To gain a better understanding of instruction-following outcomes, we analyze the internal state of LLMs, focusing on the differences in representations between success and failure cases of

---

[*] Work done while at Apple.
[1] Code and data are available at https://github.com/apple/ml-internal-llms-instruction-following

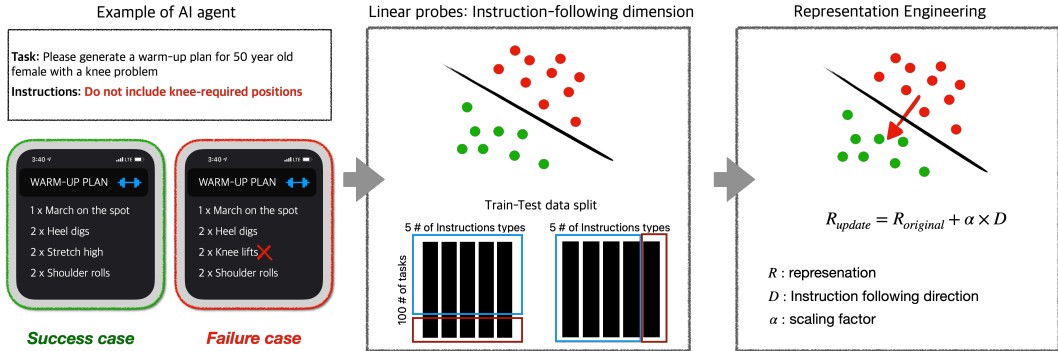

Figure 1: Overview of our paper. **Left**: Success and failure cases in a personalized AI fitness planner. The task is to generate a warm-up plan while avoiding knee-required positions. The success case follows the instruction, while the failure case violates it. **Middle**: Linear probing is applied to analyze internal representations from success and failure cases, identifying the instruction-following dimension. The probe is tested on unseen tasks (e.g., writing a CV) and instruction types (e.g., include/exclude keywords). **Right**: Representation engineering is used to shift failure cases into success by adjusting the representations along the instruction-following dimension, improving adherence without compromising task quality.

instruction-following across different tokens and layers. Our approach involves disentangling the effects of tasks and instructions in input prompts, where the *instruction* specifies the action (e.g., 'please do not use keywords') and the *task* provides the context for executing the instruction (e.g., 'please write a resume'). By applying linear probing—a widely used method for interpreting model representations (Alain & Bengio, 2016; Belinkov, 2022; Elazar et al., 2021)—we identify a specific dimension within the input embedding space that is strongly associated with instruction-following. While previous work has primarily used linear probing to explore representations related to truthfulness and reducing hallucinations (Azaria & Mitchell, 2023; Marks & Tegmark, 2023; MacDiarmid et al., 2024), our study extends this method to investigate instruction-following. We demonstrate that this dimension generalizes to unseen tasks, however not to unseen instruction types.

To validate the significance of the instruction-following dimension, we applied representation engineering techniques to enforce instruction-following based on insights from our linear probes. Our experiments show that adjustments along this specific dimension are more effective in enhancing instruction-following success rates than random modifications, while maintaining the overall quality of the generated responses. These results indicate that the instruction-following dimension plays a crucial role in shaping the model's behavior, toward better adherence to instructions.

To further interpret the meaning of this dimension, we conduct a sensitivity analysis based on three key perturbations to the input prompt: task familiarity, instruction difficulty, and phrasing. Our findings reveal that this dimension is more related to the rephrasing of prompts rather than the inherent difficulty of the task or instructions. This suggest that the way a prompt is encoded within the model's input representation space plays a significant role in whether the instruction is followed correctly. This observation not only provides a deeper understanding of why LLMs sometimes fail to adhere to straightforward instructions but also offers an explanation for the effectiveness of prompt engineering, even when the content of the prompt remains largely unchanged.

Overall, this work sheds light on the underlying mechanisms of instruction-following in LLMs by uncovering a critical dimension in the model's representation space. These insights enhance our understanding of LLM behavior and offer practical approaches to improving instruction adherence, bringing us closer to developing more reliable and trustworthy AI agents.

## 1.1 CONTRIBUTIONS

- We identify a specific dimension within the input embeddings space of LLMs that is closely linked to instruction-following, using linear probes, by carefully designing our setting to disentangle the effects of tasks and instructions in input prompts.

- We demonstrate that this dimension generalizes to unseen tasks and that modifying representations along this dimension effectively converts instruction-following failures into successes without compromising response quality.
- Through a sensitivity analysis, our findings reveal that this dimension is linked to how prompts are rephrased, underscoring that instruction-following in LLMs is influenced by how prompts are encoded within the model's input embeddings. This explains why LLMs sometimes fail to follow clear, simple instructions and why prompt engineering can enhance instruction adherence, even when the content remains largely unchanged.

## 2 DO LLMS KNOW WHEN THEY SUCCEED OR FAIL TO FOLLOW INSTRUCTIONS?

In this section, we aim to identify the dimension within the models' representation space that is closely associated with instruction-following. We use linear probes to determine the internal signals that separate successful instruction-following from failures and examine whether this dimension generalizes to different tasks and instruction types. By exploring different tokens and layers within the models, we seek to understand how and when instruction-following information is encoded.

### 2.1 IFEVAL-SIMPLE

To objectively evaluate LLMs with simple and verifiable instructions, we select IFEval (Zhou et al., 2023) as our base dataset. The motivation is that, while complex and multi-purpose instruction prompts are more realistic, they require using LLM-based evaluators that may induce further errors and biases in assessing success or failure. To avoid this potential issue, we focus on simple, single-purpose and verifiable instructions from IFEval, such as "Please do not include keywords: ..." or "answer in lower-case only", that can be automatically validated with deterministic programs like string-matching, thereby minimizing uncertainties from ambiguous evaluation criteria. We provide a more detailed justification in Appendix A.6.

The IFEval dataset comprises 25 instruction types under 9 categories, with each instruction type paired with a distinct set of tasks — approximately 20 tasks per instruction type. Furthermore, due to the relatively small number of tasks per instruction type, internal model states resulting from these prompts contain a mix of both instruction-following and task-specific details. To isolate the dimension related specifically to instruction-following, we generated a modified version of the IFEval data, called IFEval-simple.[2] First, we selected 5 instruction types that are likely to be used in real-world applications for AI agents. For example, ensuring the inclusion (keywords:existence) or exclusion (keywords:forbidden) of specific keywords, specifying the frequency of certain keywords (keywords:frequency), generating responses with placeholders (detectable_content:place_holders), and requiring responses to end with predefined sentences (startend:end checker). We excluded more complex or impractical instructions, such as those requiring omission of punctuation, as they are less relevant for practical use cases.

Second, we generated 100 tasks using GPT-4, similar to the original tasks in IFEval, where each instruction type is paired with the same set of 100 tasks. By pairing each instruction type with the same set of 100 tasks, we ensure that linear probes trained on the model's representations are more likely to capture information solely related to instruction-following, without the confounding influence of varying tasks. The instructions assigned to each task vary in detail based on the context. For example, for an instruction type focused on keyword inclusion or exclusion, a resume-writing task might require keywords like 'skills' and 'career', while a joke about a programmer might involve terms like 'syntax' or 'code'. These variations introduce diverse challenges, testing the model's adaptability in following instructions. Example tasks are provided in Appendix Table 5 and Table 6. The instruction-following accuracy for IFEval-simple datasets is presented in Appendix Table 11.

### 2.2 METHODS

**Representations** We analyzed four language models: LLaMA-2-7B-chat (Touvron et al., 2023), LLaMA-2-13B-chat (Touvron et al., 2023), Mistral-7B-Instruct-v0.3 (Jiang et al., 2023), and Phi-

---

[2]The IFEval-simple data is available at https://github.com/apple/ml-internal-llms-instruction-following.

| | Task generalization | | | Instruction-type generalization | | |
|---|---|---|---|---|---|---|
| Model | First token | Middle token | Last token | First token | Middle token | Last token |
| LLaMA-2-chat-7B (14 lyr) | $0.77 \pm 0.04$ | $0.55 \pm 0.07$ | $0.73 \pm 0.04$ | $0.52 \pm 0.03$ | $0.50 \pm 0.07$ | $0.52 \pm 0.05$ |
| LLaMA-2-chat-13B (16 lyr) | $0.83 \pm 0.03$ | $0.58 \pm 0.06$ | $0.82 \pm 0.03$ | $0.56 \pm 0.06$ | $0.58 \pm 0.06$ | $0.53 \pm 0.03$ |
| Mistral-7B-inst-v0.3 (14 lyr) | $0.74 \pm 0.02$ | $0.54 \pm 0.05$ | $0.72 \pm 0.04$ | $0.50 \pm 0.05$ | $0.51 \pm 0.05$ | $0.51 \pm 0.05$ |
| Phi-3-mini-128k (14 lyr) | $0.88 \pm 0.03$ | $0.56 \pm 0.04$ | $0.86 \pm 0.03$ | $0.55 \pm 0.04$ | $0.48 \pm 0.03$ | $0.50 \pm 0.03$ |

Table 1: **Task and instruction-type generalization** AUROC scores for task and instruction-type generalization using a 70-30 train-test split for task generalization on unseen tasks, and leave-one-out cross-validation for instruction-type generalization across different instruction types. Standard deviation is calculated from five runs with different random seeds for task generalization and across instruction types for instruction-type generalization.

| | Early layers | | | Middle layers | | | Last layers | | |
|---|---|---|---|---|---|---|---|---|---|
| Model | First token | Middle token | Last token | First token | Middle token | Last token | First token | Middle token | Last token |
| LLaMA-2-chat-7B | **$0.77 \pm 0.04$** | $0.55 \pm 0.07$ | $0.73 \pm 0.04$ | $0.75 \pm 0.05$ | $0.51 \pm 0.04$ | $0.76 \pm 0.04$ | $0.73 \pm 0.03$ | $0.54 \pm 0.02$ | $0.70 \pm 0.02$ |
| LLaMA-2-chat-13B | **$0.83 \pm 0.03$** | $0.58 \pm 0.06$ | $0.82 \pm 0.03$ | $0.81 \pm 0.02$ | $0.56 \pm 0.05$ | $0.80 \pm 0.04$ | $0.78 \pm 0.04$ | $0.49 \pm 0.03$ | $0.79 \pm 0.05$ |
| Mistral-7B-inst-v0.3 | **$0.74 \pm 0.02$** | $0.54 \pm 0.05$ | $0.72 \pm 0.04$ | $0.71 \pm 0.05$ | $0.51 \pm 0.03$ | $0.67 \pm 0.04$ | $0.71 \pm 0.03$ | $0.49 \pm 0.04$ | $0.70 \pm 0.03$ |
| Phi-3-mini-128k | **$0.88 \pm 0.03$** | $0.56 \pm 0.04$ | $0.86 \pm 0.03$ | $0.85 \pm 0.03$ | $0.56 \pm 0.03$ | $0.83 \pm 0.02$ | $0.65 \pm 0.05$ | $0.53 \pm 0.03$ | $0.63 \pm 0.04$ |

Table 2: **Task generalization (detailed across layers)** AUROC scores for the first, middle, and last tokens across early, middle, and last layers of various models. The layers selected for LLaMA-2-13B-chat are 16, 32, and 40, while for the other three models, the layers used are 14, 26, and 32.

3-mini-128k-instruct (Abdin et al., 2024). For each model, we looked at the representations on three tokens: (1) first token, $LLM(x_1, x_2, \ldots, x_n)$, where $x_i$ are the $n$ tokens in the input prompt; (2) middle token, $LLM(x_1, x_2, \ldots, x_n, y_1, y_2, \ldots, y_{m/2})$, where $y_j$ are the first $m/2$ tokens of the response; and (3) last token, $LLM(x_1, x_2, \ldots, x_n, y_1, y_2, \ldots, y_m)$, representing the full input and response. We also examined three layers (early, middle, last) to identify where instruction-following information is encoded within the models' internal state. Specifically, we used layers 16, 32, and 40 and for LLaMA-2-13B-chat and 14, 26, and 32 for other three models. To avoid randomness in decoding, we employed greedy decoding without sampling.

**Linear Probes** We trained linear probes on the representations to identify the instruction-following dimension. A simple linear model was trained on instruction-following success outcome, optimized for 1000 epochs with AdamW, a 0.001 learning rate, and 0.1 weight decay.

**Train-test split and metric** We assessed task generalization and instruction-type generalization by splitting the data into training and testing sets, as shown in Figure 1. IFEval-simple has 5 instruction types, each paired with the same set of 100 tasks. To evaluate task generalization, we split the data by the task dimension, using a 70-30 train-test split across the 100 tasks. To evaluate instruction-type generalization, we applied a leave-one-out approach, over the instruction-type dimension. To evaluate performance, we use the Area Under the Receiver Operating Characteristic Curve (AUC)(Pedregosa et al., 2011), assessing the accuracy of binary predictions for each model on unseen tasks and instruction types.

## 2.3 RESULTS

**Linear probes generalize across unseen tasks** The task generalization results in Table 1 show that linear probes performed well across different tasks when the instruction type remains consistent. The AUROC scores, which range from 0.7 to 0.8 using the first token, suggest that the input embeddings of these models possess a shared geometry related to instruction-following that generalizes well across varied tasks. This is particularly beneficial in the context of buliding AI agents, where a pre-defined consistent set of instructions needs to be followed across different tasks. For example, if a probe is trained on examples of an instruction type like "Please do not include these keywords" using examples from resume writing and nutrition coaching, the linear probe can predict if the model follows the same instructions type even unseen tasks, such as creating a warm-up plan without knee-intensive exercises. Additionally, we plot the principal components analysis (PCA) using representations from the first token and early layers, fitting the PCA on the training split and visualizing the results on the test split (unseen tasks) in Figure 2. They show clear separability, sup-

| Instructions | LLaMA-2-chat-7B | | | LLaMA-2-chat-13B | | | Mistral-7B-inst-v0.3 | | | Phi-3-mini-128k | | |
|---|---|---|---|---|---|---|---|---|---|---|---|---|
| | Early lyr | Middle lyr | Last lyr | Early lyr | Middle lyr | Last lyr | Early lyr | Middle lyr | Last lyr | Early lyr | Middle lyr | Last lyr |
| key:forbidden | 0.52 | 0.51 | 0.56 | 0.45 | 0.45 | 0.44 | 0.44 | 0.41 | 0.46 | 0.52 | 0.54 | 0.53 |
| key:exist | 0.50 | 0.50 | 0.51 | 0.67 | 0.68 | 0.66 | 0.55 | 0.50 | 0.50 | 0.63 | 0.67 | 0.68 |
| key:freq | 0.57 | 0.59 | 0.59 | 0.57 | 0.57 | 0.57 | 0.56 | 0.56 | 0.56 | - | - | - |
| number_placeholders | 0.56 | 0.54 | 0.52 | 0.58 | 0.58 | 0.54 | 0.50 | 0.49 | 0.50 | 0.50 | 0.53 | 0.46 |
| end_checker | 0.48 | 0.46 | 0.47 | 0.55 | 0.57 | 0.56 | 0.44 | 0.42 | 0.45 | 0.55 | 0.59 | 0.57 |
| AVERAGE | 0.52 | 0.52 | 0.53 | 0.56 | 0.57 | 0.55 | 0.50 | 0.48 | 0.49 | 0.55 | 0.58 | 0.56 |

Table 3: **Instruction-type generalization (detailed)** AUROC across different models and selected layers on first token representations. A leave-one-out approach was employed, and the standard deviation from training a linear probe is small enough to be omitted from the table. The '-' mark in 'keywords:frequency' instruction type is due to an insufficient number of data points caused by a 100% success rate, making it impossible to compute reliable AUC scores.

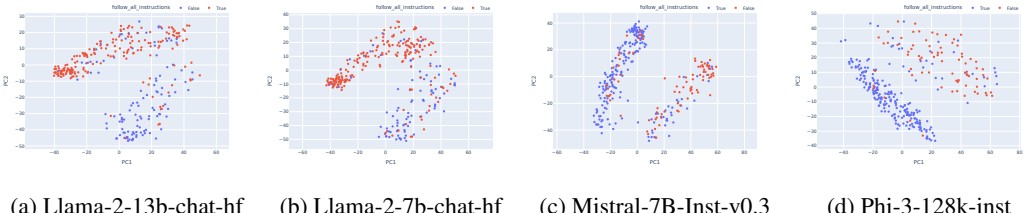

(a) Llama-2-13b-chat-hf (b) Llama-2-7b-chat-hf (c) Mistral-7B-Inst-v0.3 (d) Phi-3-128k-inst

Figure 2: **PCA plot** of first token representations from early layers across four LLMs. PCA is fitted on the training split and visualized on the test split (unseen tasks). The PCA shows separability, suggesting the consistent capture of the instruction-following dimension across tasks. The analysis includes three instruction types from the keyword category in IFEval-simple. Additional PCA results for all five instruction types across different categories are provided in Appendix Figure 6.

porting the idea that the instruction-following dimension is consistently represented across different tasks. Further PCA analysis is provided in Figure 6 in the Appendix.

**Linear probes do not generalize across unseen instruction types** In contrast to task generalization, the models exhibit no clear generalization when tested across unseen instruction types. The AUROC scores for instruction-type generalization are notably lower, ranging from 0.50 to 0.55, close to chance (Table 1). A potential explanation for this poor generalization could be the limited number of instruction types used during training, where the linear probe was trained on just 4 instruction types. To investigate, we expanded the dataset to include 25 instruction types, each paired with 20 tasks. However, as shown in Appendix in Table 8, this expanded experiment yielded similar results, with models still failing to generalize well across unseen instruction types. This indicates that models struggle to generalize instruction-following across different instruction types, implying the absence of a 'global' instruction-following dimension that can be leveraged regardless of the instruction type, which may be due to varying representation geometries.

**First token is as informative as last token** Interestingly, the first and last tokens—representing the model's state before and after response generation—show high AUROC scores, implying that LLMs may already "know" whether they will follow instructions even before they start generating their responses. This early indication of instruction following is valuable, since early intervention or correction could be applied. In contrast, the middle tokens showed lower AUROC scores, likely because the representation contains information about next token generation more than information about instruction-following.

**Layer-wise performance is similar, with early layers slightly better for task generalization** The performance across different layers shows only slight variations, with early layers marginally outperforming middle and last layers, as detailed in Table 2. For example, in the 13B model, the early layers achieve an AUROC of 0.83 for the early token, which is slightly better than the performance of middle and last layers. This suggests that the instruction-following dimension may be more prominently represented in the earlier stages of the model's processing. However, for instruction-type generalization, there is no clear pattern across layers (Table 3), indicating that the challenges associated with generalizing across different instruction types are pervasive throughout layers.

| Model | Original SR | Random SR | Inst-follow SR | Original QR | Random QR | Inst-follow QR |
|---|---|---|---|---|---|---|
| LLaMA-2-chat-7B | $0.57 \pm 0.00$ | $0.55 \pm 0.00$ | $\mathbf{0.59 \pm 0.00}$ | $0.87 \pm 0.09$ | $0.85 \pm 0.10$ | $0.87 \pm 0.08$ |
| LLaMA-2-chat-13B | $0.61 \pm 0.00$ | $0.54 \pm 0.12$ | $\mathbf{0.65 \pm 0.02}$ | $0.92 \pm 0.00$ | $0.91 \pm 0.02$ | $\mathbf{0.94 \pm 0.00}$ |
| Mistral-7B-inst-v0.3 | $0.58 \pm 0.00$ | $0.56 \pm 0.02$ | $\mathbf{0.64 \pm 0.02}$ | $0.95 \pm 0.02$ | $0.86 \pm 0.02$ | $0.98 \pm 0.06$ |
| Phi-3-mini-128k | $0.71 \pm 0.00$ | $0.63 \pm 0.04$ | $\mathbf{0.74 \pm 0.01}$ | $0.76 \pm 0.01$ | $0.76 \pm 0.01$ | $\mathbf{0.78 \pm 0.00}$ |

Table 4: **Representation Engineering results on the last layer across four models**. Success rate (SR) for instruction-following and quality ratio (QR) for task quality are compared across the original outputs, outputs using the instruction-following dimension, and outputs using a random directions. RE along the instruction-following dimension improves SR while maintaining or enhancing QR, unlike random adjustments which often reduce both SR and QR. Standard deviations are across three runs with different random seeds.

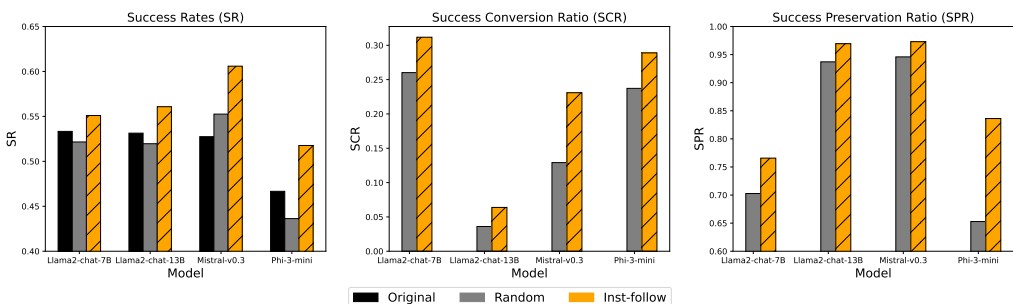

Figure 3: **Transition metric for Representation Engineering on the last layer of four models** Success rate (SR) only on high quality responses in task execution (scoring above 7 by GPT-4, scale from 0 to 9). The Success conversion ratio (SCR) indicates the proportion of originally failed responses that became successful after modification, while Success preservation ratio (SPR) reflects the proportion of originally successful responses that remained successful.

# 3 REPRESENTATION ENGINEERING

We identified a dimension within the input embedding space associated with instruction-following. To evaluate whether this dimension significantly impacts the models' behavior, we manipulated the representations along this direction using representation engineering (Marks & Tegmark, 2023; Zou et al., 2023). An increase in the models' instruction-following success rate tied to manipulations along the identified direction validates the role of the dimension in shaping the models' generation outcomes toward instruction adherence.

## 3.1 SETTINGS

**Method** For each input representation $R_{original}$, we applied a transformation in the identified direction $D$ using the formula $R_{updated} = R_{original} + \alpha \times D$, where $\alpha$ is a scaling hyper-parameter. We applied this transformation to all input representations, including both success and failure cases, to evaluate whether RE could improve instruction following universally, without disrupting cases where the model was already successful. This adjustment was applied to the representations in the last layer of the model, as it was more robust to variations in $\alpha$. We focused on the representation of the first token, which corresponds to the input embedding before any response generation, since the goal of representation engineering (RE) is to adjust internal representations before the response is generated to improve the model's instruction adherence. The direction $D$ is the weight of a linear probes trained on all IFEval-simple dataset. [3]

**Metric** We evaluated the success rate (SR) of instruction-following using predefined evaluation functions from the IFEval (Zhou et al., 2023). Additionally, we assessed the quality of the responses

---

[3]We also experimented with training the linear probe on 70% of the IFEval-simple dataset and applying RE to the remaining 30% test set. The results were similar but slightly worse than when the linear probe was trained and RE was applied to the entire dataset. Since our primary focus is on analyzing the variance caused by RE itself, rather than variance from train-test splits, we present the results using the full dataset here.

using GPT-4, scoring each response on a scale from 0 to 9 based on its relevance to the given task. We defined quality ratio (QR) as the number of responses scoring above 7 divided by the total number of responses that successfully follow instructions (this cutoff was defined based on the distribution of quality scores). F2T (False to True) and T2T (True to True) show how many failed responses became successful and how many successful ones remained so after modification. The Success conversion ratio (SCR) $:= \frac{F2T}{(F2T+F2F)}$ indicates the proportion of originally failed responses that became successful after modification, while Success preservation ratio (SPR) $:= \frac{T2T}{(T2T+T2F)}$ reflects the proportion of originally successful responses that remained successful.

**Baseline and hyperparameter selection** To demonstrate the effectiveness of the identified instruction-following dimension, we compared it against random directions. Each model and instruction type required a different $\alpha$ value based on their specific geometry. If $\alpha$ is too large, it can degrade the quality of responses; if too small, it may not effectively improve instruction-following. We selected $\alpha$ for each model and instruction type using a validation set comprising 10% of the instruction data. The selected $\alpha$ values were: 0.3 for Llama-2-chat-13b and Llama-2-chat-7b, 0.1 for Phi-3, and 0.15 for Mistral-7B.

---

**Prompt for scoring task quality**

You are a helpful assistant in evaluating the quality of the outputs for a given instruction. Your goal is to score a given output for the given instruction. You should give an overall score (an integer) on a scale of 0 to 9, where a higher score indicates better overall performance. Do NOT provide any explanation for your evaluation.

# Instruction: {Task-only-input}
# Output:{Response}
# Score of the Output (Your response should be ONLY the score, an integer between 0-9):

---

## 3.2 RESULTS

**RE on instruction-following direction improves success rate while maintaining quality** Our experiments demonstrate that applying the RE direction generally improves the instruction-following success rate (SR) across most models and instruction types. As shown in Table 4, the SR with the instruction-following direction usually outperforms the original success rate and is lower bounded by the the original SR – that is, the instruction-following dimension does not lead to worse than original SRs. Additionally, the QR remains equal to or higher than the original, indicating that RE can be applied with minimal risk of reducing response quality. Figure 5 in the Appendix provides an illustrative example of modified responses. In this case, the task was to write a resume with the instruction to include three specific keywords. The original response only included one keyword, whereas the modified response, guided by the instruction-following direction, successfully incorporated all three keywords, demonstrating the effectiveness of RE in enhancing instruction adherence.

**Instruction-following direction is better than random directions** When comparing RE direction to random directions, RE consistently outperforms random directions in increasing the success rate across all instruction types and models, as illustrated in Table 4 and Figure 3. The ratios of True-to-True (T2T) and False-to-True (F2T) transitions are typically larger for the instruction-following direction than for random directions, indicating a more reliable improvement in success rates.

## 4 INTERPRETING THE INSTRUCTION-FOLLOWING DIMENSION

While manipulating representations along the instruction-following dimension reveals that it influences a model's behavior, the meaning behind this manipulation remains unclear. To interpret the meaning of the instruction-following dimension, we conduct a sensitivity analysis to investigate the relative of perturbations on the internal state of LLMs, compared to our identified direction. We consider three perturbation types: task familiarity, instruction difficulty, and phrasing. We (1) systematically alter the original input prompts in IFEval-simple dataset for each perturbation, (2) compute the resulting difference in internal state representation space before and after the perturbation,

and (3) compute the cosine similarity between the perturbation-induced difference vector and the instruction-following dimension we identified. We designed prompt changes for each perturbation:

**(1) Task Familiarity:** We investigated whether the instruction-following dimension might be related to how familiar the model is with a given task. For example, the task "Write a resume for software engineer" might be more familiar to the model than "Write a summary about current events", if it was more common in the data used to train the LLMs. If a task is more familiar to a model, it may be easier for the model to follow instructions regarding that task. To perturb the model on task familiarity, we kept the instruction constant while changing the task to one with lower perplexity (Jelinek et al., 1977). Perplexity measures the probability of tokens in generation, reflecting task familiarity (Gonen et al., 2022), where high perplexity indicates a familiar task and vice versa.

**(2) Instruction Difficulty:** We investigated the relationship of the instruction-following dimension with the complexity of the instructions. We perturbed the instruction difficulty by simplifying instructions by relaxing instruction-related constraints. For example, in the original instruction "Please include keywords: coding, Python, computer, experience", we reduced the complexity by reducing the number of keywords required in the instruction to "Please include the keywords: coding".

**(3) Phrasing Modification:** Finally, we examined whether the instruction-following dimension was correlated to how the prompt is phrased. We rephrased the prompts while keeping the meaning of the task and the instruction unchanged. For example, we modified "Write a resume for software engineer. Please include keywords such as coding, Python, computer, experience" to "I want you to write about software engineer resume including four words coding, Python, computer, or experience". We used GPT-4 to rephrase both the task and instruction in the input prompt, and applied GPT-4 again to validate that the meaning of the contents remained the same after rephrasing.

We selected 20 prompts, each containing a task and an instruction from the 'forbidden keyword' instruction type in IFEval-simple dataset. For each perturbation type, we created five modified versions of each prompt. We then averaged the representations of these modified prompts and calculated the difference between this averaged representation and the representation of the original prompt. Finally, we assessed how well this difference vector aligned with the instruction-following dimension by computing the cosine similarity.

Our findings, illustrated in Figure 4, show the sensitivy analysis results for two models: Llama-2-13b-chat and Llama-2-7b-chat. In both models, the results indicated that phrasing modifications have a stronger correlation with the instruction-following dimension than task familiarity or instruction difficulty. These results support the hypothesis that the instruction-following dimension is more closely tied to how prompts are phrased rather than the inherent difficulty of the task or the complexity of the instruction. This suggests that how prompts are phrased plays a critical role in determining whether LLMs will successfully follow the instructions, aligned to observations Lu et al. (2023); Sclar et al. (2023) showing LLMs are sensitive to prompt formatting.

## 5 RELATED WORK

**Instruction-following in LLMs** Recent research has introduced various benchmark datasets to evaluate the instruction-following capabilities of LLMs across different contexts(Zhou et al., 2023; Qin et al., 2024; Yan et al., 2024; Xia et al., 2024). Beyond evaluation, several approaches have been proposed to improve instruction-following performance, such as modifying attention mechanisms (Zhang et al., 2023) and applying fine-tuning strategies (He et al., 2024; Sun et al., 2024). In contrast to prior work that primarily focuses on evaluating or enhancing instruction-following, our study aims to understand *why LLMs sometimes fail to follow instructions by analyzing internal representations*.

**Linear Probing and Representation engineering on LLMs** Linear probes have been widely used for interpreting and analyzing the representations of neural networks (Alain & Bengio, 2016) and language models (Belinkov, 2022; Elazar et al., 2021). Specifically, probing for the trustworthiness of LLMs has been an active area of research (Azaria & Mitchell, 2023; Marks & Tegmark, 2023; MacDiarmid et al., 2024; Li et al., 2024a; Burns et al., 2022; Zou et al., 2023; Rimsky et al., 2023; Li et al., 2022; Nanda et al., 2023; Subramani et al., 2022; Tigges et al., 2023; Todd et al., 2023; Farquhar et al., 2024; Ahdritz et al., 2024; Duan et al., 2024). These probing methods are closely related to representation engineering and editing techniques aimed at modifying model knowledge

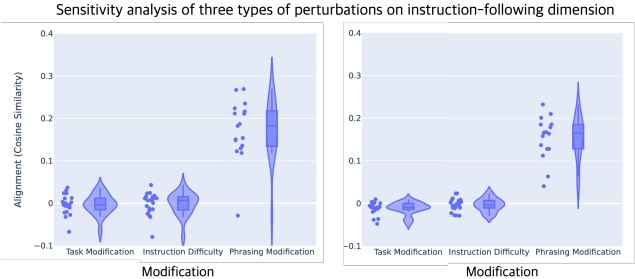

Figure 4: **Cosine similarity alignment for modified data** in the 'forbidden keyword' instruction type across two models (Llama-2-7b-chat (**Left**) and Llama-2-13b-chat (**Right**)). The figure shows the cosine similarity between the instruction-following dimension and the difference vector (computed as the difference between the original prompt's representation and the average representation of five modified prompts) across 20 sampled prompts. Modifications include changes in task familiarity, instruction difficulty, and phrasing. The results indicate that phrasing modifications align more closely with the instruction-following dimension, suggesting that how prompts are phrased plays a crucial role in determining instruction adherence.

and behavior (Zou et al., 2023; Rimsky et al., 2023; Li et al., 2024a; Park et al., 2023; Chen & Yang, 2023; Luo et al., 2024; Turner et al., 2023). Our work is distinct from these previous efforts, which primarily focus on representations related to truthfulness and reducing hallucinations. In contrast, our study centers on representations related to instruction-following, highlighting the importance of understanding how models internally handle instructions.

## 6  DISCUSSION AND CONCLUSION

### 6.1  LLMS INTERNALLY RECOGNIZE WHETHER THEY WILL FOLLOW INSTRUCTIONS

Our findings suggest that LLMs may possess an inherent ability to predict whether they will successfully follow instructions, even before the generation process begins . This capability is supported by several key observations:

**LLMs generalize well across tasks but struggle with different instruction types** We find that while LLMs can generalize across different tasks, they struggle with generalization across different instruction types. This suggests that distinct instruction categories may have unique geometries within the models' internal representation space, making it more challenging to generalize across them.

**LLMs can predict instruction success from the first token** We observe that the model's internal representations are separable from the very first token, which corresponds to the embedding of the input prompt. This indicates that the likelihood of instruction-following success can be determined early in the process, before the model generates any responses. This highlights the critical role of how the input prompt is encoded and the importance of input representations in predicting instruction-following outcomes.

**Representation engineering increases instruction-following success** We further validate the significance of the identified instruction-following dimension by adjusting the model's representations. By moving failure cases into the success class along this dimension and comparing the results to random adjustments, we observe a significant increase in the success rate while keeping the task quality. This demonstrates that the identified dimension is both meaningful and can be used practically.

**The instruction-following dimension is closely tied to prompt phrasing** Our findings, in Figure 4, reveal that the instruction-following dimension is most closely associated with the phrasing of prompts, rather than the inherent difficulty of the task or the specific details of the instructions. This suggests that how instructions are phrased plays a crucial role in whether LLMs will follow them and is consistent with our finding on the separability of representations from the early token.

## 6.2 THE ROLE OF INPUT PROMPT REPRESENTATION IN INSTRUCTION-FOLLOWING FAILURES

Our findings highlight the role of representation of the input prompt in determining instruction-following success in LLMs. We discover that the instruction-following dimension identified in our analysis is sensitive to changes in how the input prompt is phrased. This sensitivity explains several behaviors of LLMs:

**Why LLMs fail in following instructions** LLMs may fail to follow even simple, clear instructions because the encoding of the input prompt within the models' internal representation space can be easily disrupted. Our findings suggest that small variations in how a prompt is phrased can result in significant differences in how the model processes the instruction, leading to failures in adherence. This issue arises not from ambiguity in the instruction itself, but from the LLM's sensitivity to the exact structure and phrasing of the input, which influences how the instruction is embedded and processed internally. As a result, the model might not consistently follow instructions, even when they are clear and familiar.

**Why Prompt Engineering (PE) works** PE operates by slightly altering the phrasing of a prompt, which in turn changes how the input is encoded within the model. This subtle shift in encoding can move a representation from a failure class to a success class in terms of instruction-following within the input embedding space. Our work with representation engineering achieves a similar outcome, but instead of modifying the input text, we make adjustments directly in the representation space. Both approaches influence the model's internal states, highlighting the importance of the input encoding process. Our observations align with prior research showing LLM sensitivity to prompt formatting (Lu et al., 2023; Sclar et al., 2023; Gonen et al., 2022).

**Semantic sensitivity of LLM input embedding space** The fact that instruction-following success or failure can be altered by slight prompt rephrasing shows that the LLM's input embedding space is semantically sensitive. This sensitivity suggests that the model's internal representation of prompts is brittle, making LLMs vulnerable to small changes in how an input is framed or phrased. This fragility, likely driven by the model's large size and the complexity of its training dynamics, creates challenges in ensuring robust instruction adherence. Given this sensitivity, future efforts should focus on making LLMs' input embedding space more robust and reliable. One potential approach is to fine-tune models with an explicit focus on stabilizing instruction-following by utilizing the identified instruction-following dimension.

Our findings highlight the crucial role of prompt encoding in instruction-following success for LLMs. The sensitivity of the input embedding space to slight changes in phrasing explains why LLMs may fail to follow even clear instructions and why prompt engineering is effective. By adjusting the representations directly, as we did with representation engineering, we show that it is possible to significantly improve instruction adherence. Going forward, improving the robustness of LLMs' input embeddings through training can make models more reliable and consistent in following instructions across a variety of tasks. This is crucial for building trustworthy AI systems, especially in real-world applications where accuracy and reliability are essential.

## 6.3 LIMITATIONS AND FUTURE WORK

Our analysis was primarily focused on a specific set of tasks and models. Although our current results are consistent across the models we studied, future work could extend these findings by evaluating additional models to determine whether the identified instruction-following dimension generalizes across different LLM architectures. Additionally, expanding the dataset to include a wider variety of instruction-following cases could enrich the analysis and improve the generalizability of our findings. We focused our investigation on simple modeling approaches to identify an instruction-following dimension and evaluate its practical significance. Future work could include additional methods train linear probes, particularly in handling domain shifts. Similarly, better approaches to representation engineering (Zou et al., 2023) could further improve the success rate of instruction-following modifications. Finally, unambiguously interpreting the meaning of the instruction-following dimension remains an open question. We considered three hypotheses and found that phrasing modification was most closely related to the dimension associated with instruction-following using a perturbation-based approach. Additional investigations to develop systematic approaches to interpret the dimension could add to a deeper understanding of its meaning and implications.

ACKNOWLEDGMENTS

This work was conducted during an internship at Apple AIML. We sincerely thank Fahad Kamran and Feng Zhu for their valuable feedback and insightful suggestions on this work. We are also grateful to Guillermo Sapiro for his unwavering support and guidance throughout the research.

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

# A APPENDIX

## A.1 EXAMPLES OF IFEVAL-SIMPLE DATASET

The IFEval-simple dataset is created to focus specifically on instruction-following, removing the confounding influence of varying tasks present in the IFEval dataset (Zhou et al., 2023). In this modified version, we select 5 instruction types commonly used in real-world AI applications, such as including or excluding keywords, generating responses with placeholders, and ensuring specific phrases are present in the generated text. These instructions are paired with the same set of 100 tasks to help isolate the instruction-following dimension. By using the same set of tasks across all instruction types, we ensure that any differences in model behavior are attributed to instruction-following rather than task-specific features. This allows us to more effectively probe the model's internal representations and evaluate how well it can follow instructions across various scenarios.

Table 5 presents examples from the IFEval-simple dataset, such as tasks like writing a resume or creating a joke about programmers. The instructions assigned to each task vary, requiring the model to follow specific guidelines such as including or excluding certain keywords, ensuring word usage meets a specific frequency, and adhering to formatting rules. The keywords that must be included or excluded differ based on the task. For instance, in the resume task, keywords might include "resume", "software", or "engineer", whereas in the joke task, the focus may shift to terms like "syntax" or "code". These varied instructions introduce diverse challenges for the model in instruction-following.

| Type | Example |
|---|---|
| **Task** | *Write a resume for a software engineer with 5+ years of experience in the Bay Area, CA.* |
| **Instruction** | **keywords:existence**
Make sure to include the keywords: "skills", "technology", "career".

**keywords:forbidden**
Do not include the following keywords: resume, software, engineer, experience.

**keywords:frequency**
Make sure to use the word "qualifications" at least 2 times.

**startend:end checker**
Your resume must end with the exact phrase "Looking forward to contributing to innovative projects."

**detectable content:number placeholders**
Make sure to include at least 5 placeholders represented by square brackets, such as [name]. |
| **Task** | *Write a joke about programmers.* |
| **Instruction** | **keywords:existence**
Make sure to include the keywords: "humor", "code", "life".

**keywords:forbidden**
Do not include the following keywords: joke, programmers.

**keywords:frequency**
Make sure to use the word "syntax" at least 3 times.

**startend:end checker**
Your programmer joke must end with the exact phrase "And that's the real bug in the code of life."

**detectable content:number placeholders**
Make sure to include at least 3 placeholders represented by square brackets, such as [name]. |

Table 5: **Examples from the IFEval-simple dataset.** This table shows two tasks: writing a resume and crafting a joke about programmers. Each task is paired with multiple instruction types, such as including/excluding keywords, ensuring word frequency, and adhering to specific content formatting rules. The uniform set of tasks across different instruction types helps isolate the instruction-following dimension by removing task-specific variations.

| Index | Task |
|---|---|
| 1 | Write a story about the importance of understanding the truths that are not obvious. |
| 2 | Write a serious riddle about trips and stitches in a poem style. |
| 3 | Write a rubric for teenagers on how to review a book. |
| 4 | Write a persuasive email to a teenager who lives in Aberdeen, Scotland. |
| 5 | Write a resume for a software engineer with 5+ years of experience in the Bay Area, CA. |
| 6 | Write a song about regrets in the style of Taylor Swift. |
| 7 | Write an essay about Alvin and the Chipmunks. |
| 8 | The Legend of the Sword and the Fairy is a movie in which Wan Wan is a villain. Write a story about Wan Wan's character. |
| 9 | Write a story about a family that goes camping in the woods. |
| 10 | Write an obviously fake news article saying that aliens have invaded earth. Make it funny. |
| 11 | Write a song about the benefits of eating your vegetables. |
| 12 | Write a startup pitch for "Ward and Guerre". |
| 13 | Is Seoul a good place to live? |
| 14 | Write a letter to a friend asking them to go and vote. |
| 15 | Write a resume for a fresh high school graduate who is seeking their first job. |
| 16 | Is praying for someone's health a good idea? |
| 17 | What's the difference between a 2-stroke and a 4-stroke motor? |
| 18 | Explain to a group of elementary school students why we have seasons. |
| 19 | Can you re-create a story from a fictional newspaper with the title: "A man mysteriously died in his house, and police are investigating"? |
| 20 | Come up with a proposal for a new research project on how to improve the quality of life for people with disabilities. |
| 21 | Write a blog post about the benefits of meditation for busy professionals. |
| 22 | Create a recipe for a vegan gluten-free chocolate cake. |
| 23 | Draft a comprehensive guide on how to start a podcast. |
| 24 | Develop a character sketch for a villain in a fantasy novel. |
| 25 | Compose a haiku about a sunset over the ocean. |
| 26 | Summarize the plot of the film "Inception". |
| 27 | Explain the theory of relativity in simple terms. |
| 28 | Write a review of the latest iPhone model. |
| 29 | Describe the lifecycle of a butterfly. |
| 30 | Propose a business plan for a sustainable fashion brand. |
| 31 | Outline the steps for training a puppy. |
| 32 | Discuss the impact of social media on teenage mental health. |
| 33 | Draft a speech for a climate change conference. |
| 34 | Write a joke about programmers. |
| 35 | Explain how to change a car tire. |
| 36 | Develop a fitness routine for beginners. |
| 37 | Compose a sonnet about the city of Venice. |
| 38 | Write a user manual for a smartwatch. |
| 39 | Describe a typical day in ancient Rome. |
| 40 | Provide advice on how to improve public speaking skills. |
| 41 | Discuss the effects of global warming on polar bears. |
| 42 | Draft a letter of recommendation for a student. |
| 43 | Summarize the story of "The Great Gatsby". |
| 44 | Explain the process of photosynthesis. |
| 45 | Write a critique of a famous painting. |
| 46 | Develop a marketing strategy for a new video game. |
| 47 | Compose a limerick about a mischievous cat. |
| 48 | Describe the benefits of yoga for athletes. |
| 49 | Write instructions for assembling a desk. |
| 50 | Discuss the history of the internet. |

Table 6: **Sample of 50 tasks from the IFEval-simple dataset.** This table provides a subset of 50 tasks from the IFEval-simple dataset, which includes a total of 100 tasks designed to evaluate instruction-following performance.

## A.2    EXAMPLE OF REPRESENTATION ENGINEERING

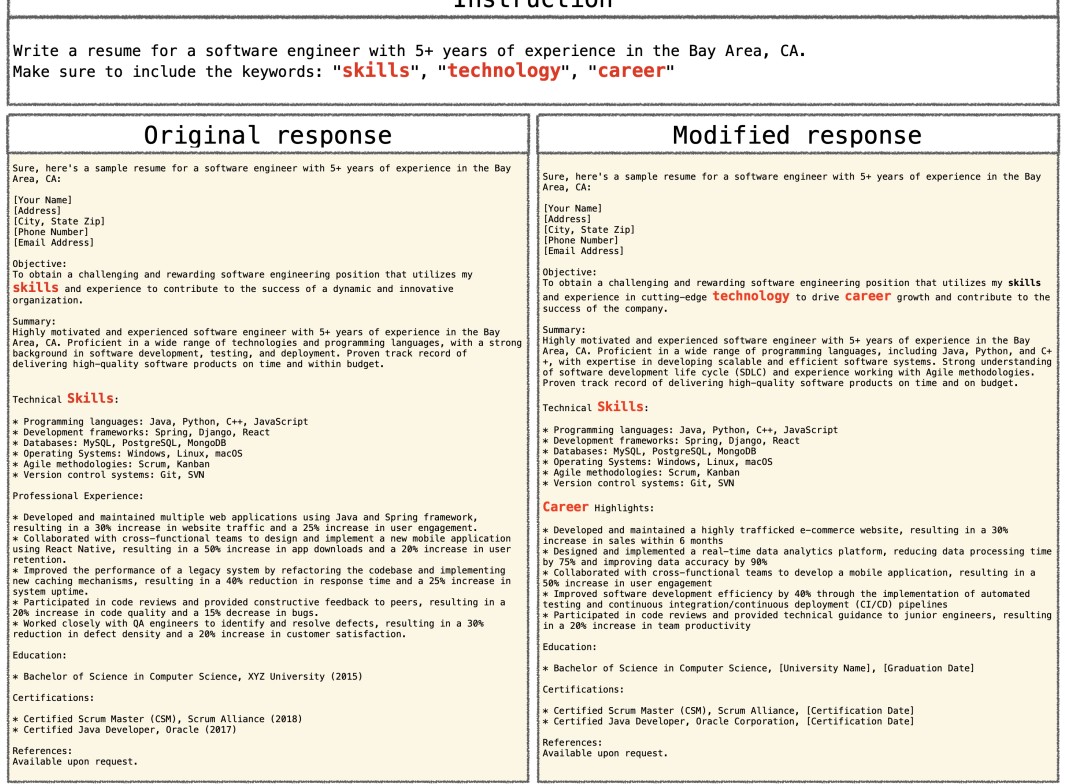

Figure 5: **RE example** An illustrative example of modified responses. In this case, the task was to write a resume with the instruction to include three specific keywords. The original response only included one keyword, whereas the modified response, guided by the instruction-following direction, successfully incorporated all three keywords, demonstrating the effectiveness of RE in enhancing instruction adherence.

## A.3    INSTRUCTION GENERALIZATION ON EXPANDED EXPERIMENT

In the main paper, we observed that models struggle to generalize across unseen instruction types, with AUC scores ranging from 0.50 to 0.55, which is close to random chance, as shown in Table 1 and Table 3 of the main paper. One hypothesis for this poor generalization is the limited number of instruction types used in the initial experiments, where the linear probe was trained on just 4 instruction types. To further investigate this, we expanded the dataset to include 23 instruction types across 8 categories, each paired with 20 tasks.

Unlike the IFEval dataset, which contains 25 instruction types across 9 categories, we omitted the 'combination' category, which includes the 'combination: Repeat Prompt' and 'combination: Two Responses' instruction types. This is because combined instructions can lead to conflicting signals in our analysis, where success in one instruction type but failure in another may produce mixed representations. By focusing on single instruction types, we aim to more clearly capture the representations associated with instruction-following success and failure. In comparison to IFEval-simple, which features 5 instruction types across 3 categories, this expanded dataset includes 23 instruction types across 8 categories, helping to prevent overfitting to a small number of instructions.

The results from this expanded experiment, shown in Table 7 for different layers and Table 8 for different tokens, reveal that despite increasing the number of instruction types, the models still demonstrate limited generalization across unseen instruction types. The AUC scores remain close to chance levels, similar to the initial experiments. As shown in Table 7 and 8, the results indicate

that adding more instruction types does not significantly improve instruction generalization. These findings reinforce the conclusion that models struggle to generalize instruction-following across different instruction types. This suggests that a "global" instruction-following dimension, applicable across diverse instruction types, may not exist.

| Models | LLaMA-2-chat-7B | | | LLaMA-2-chat-13B | | | Mistral-7B-inst-v0.3 | | | Phi-3-mini-128k | | |
|---|---|---|---|---|---|---|---|---|---|---|---|---|
| Instructions | Early lyr | Middle lyr | Last lyr | Early lyr | Middle lyr | Last lyr | Early lyr | Middle lyr | Last lyr | Early lyr | Middle lyr | Last lyr |
| startend | 0.70 | 0.61 | 0.57 | 0.47 | 0.54 | 0.52 | 0.56 | 0.62 | 0.59 | 0.60 | 0.46 | 0.48 |
| keywords | 0.39 | 0.49 | 0.48 | 0.53 | 0.46 | 0.45 | 0.42 | 0.43 | 0.45 | 0.59 | 0.48 | 0.47 |
| detectable_format | 0.52 | 0.45 | 0.42 | 0.50 | 0.47 | 0.47 | 0.49 | 0.45 | 0.41 | 0.81 | 0.79 | 0.70 |
| length_constraints | 0.40 | 0.30 | 0.33 | 0.60 | 0.50 | 0.52 | 0.44 | 0.57 | 0.56 | 0.69 | 0.52 | 0.52 |
| punctuation | - | - | - | 0.47 | 0.37 | 0.35 | 0.94 | 0.95 | 0.92 | - | - | - |
| change_case | 0.59 | 0.40 | 0.35 | 0.28 | 0.26 | 0.29 | 0.61 | 0.43 | 0.39 | 0.40 | 0.34 | 0.29 |
| detectable_content | 0.65 | 0.62 | 0.61 | 0.59 | 0.53 | 0.57 | 0.49 | 0.37 | 0.34 | 0.13 | 0.11 | 0.10 |
| language | 0.38 | 0.49 | 0.47 | 0.12 | 0.13 | 0.17 | 0.41 | 0.60 | 0.62 | 0.78 | 0.77 | 0.80 |
| **AVERAGE** | 0.52 | 0.48 | 0.46 | 0.44 | 0.41 | 0.42 | 0.54 | 0.55 | 0.54 | 0.57 | 0.50 | 0.48 |

Table 7: **Instruction-type generalization on IFEval-simple-expanded across layers** AUC scores across different models and instruction types from IFEval-simple-expanded. The 'punctuation' instruction type is marked with '-' due to an insufficient number of data points caused by a low success rate, making it impossible to compute reliable AUC scores.

| | LLaMa2-chat-7b | | | LLaMa2-chat-13b | | | Mistral-7B-inst-v0.3 | | | Phi-3-mini-128k | | |
|---|---|---|---|---|---|---|---|---|---|---|---|---|
| instructions | Early token | Middle token | Last token | Early token | Middle token | Last token | Early token | Middle token | Last token | Early token | Middle token | Last token |
| startend | 0.70 | 0.42 | 0.29 | 0.47 | 0.53 | 0.55 | 0.56 | 0.56 | 0.60 | 0.60 | 0.70 | 0.64 |
| keywords | 0.39 | 0.69 | 0.66 | 0.53 | 0.32 | 0.40 | 0.42 | 0.60 | 0.50 | 0.59 | 0.37 | 0.47 |
| detectable_format | 0.52 | 0.45 | 0.49 | 0.50 | 0.58 | 0.52 | 0.49 | 0.60 | 0.57 | 0.81 | 0.56 | 0.62 |
| length_constraints | 0.40 | 0.57 | 0.55 | 0.60 | 0.61 | 0.56 | 0.44 | 0.55 | 0.56 | 0.69 | 0.44 | 0.49 |
| punctuation | - | - | - | 0.47 | 0.47 | 0.49 | 0.94 | 0.65 | 0.43 | - | - | - |
| change_case | 0.59 | 0.52 | 0.51 | 0.28 | 0.58 | 0.45 | 0.61 | 0.47 | 0.48 | 0.40 | 0.45 | 0.37 |
| detectable_content | 0.65 | 0.53 | 0.56 | 0.59 | 0.47 | 0.55 | 0.49 | 0.54 | 0.45 | 0.13 | 0.38 | 0.33 |
| language | 0.38 | 0.46 | 0.36 | 0.12 | 0.56 | 0.51 | 0.41 | 0.59 | 0.75 | 0.78 | 0.40 | 0.46 |
| **AVERAGE** | 0.52 | 0.52 | 0.49 | 0.44 | 0.51 | 0.50 | 0.54 | 0.57 | 0.54 | 0.57 | 0.47 | 0.48 |

Table 8: **Instruction-type generalization on IFEval-simple-expanded across tokens** AUC scores across early, middle, and late token representations, showing instruction-type generalization performance on IFEval-simple-expanded. The results indicate that despite expanding the number of instruction types, models continue to struggle with unseen instruction types, with scores close to chance levels across different token positions. The 'punctuation' instruction type is marked with '-' due to an insufficient number of data points caused by a low success rate, making it impossible to compute reliable AUC scores.

## A.4 SUCCESS RATE

This section presents the success rate for instruction-following, which measures the accuracy of responses adhering to instructions. The success rates for the IFEval dataset(Zhou et al., 2023) are shown in Table 9, for our IFEval-simple dataset in Table 10, and for IFEval-simple-extended in Table 11, which is used in Section A.3 of the Appendix. The IFEval dataset consists of 25 instruction types categorized under 9 broader categories, with approximately 20 tasks per instruction type. For details on IFEval and IFEval-simple, please refer to Section 2.1 of the main paper. We use the success rate (loose) metric from Zhou et al. (2023). To ensure consistent results without randomness in decoding, we used greedy decoding without sampling when calculating the success rate.

| IFEval inst | LLaMa2-chat-7b | LLaMa2-chat-13b | Mistral-7B-inst-v0.3 | Phi-3-mini-128k |
|---|---|---|---|---|
| change_case | 0.48 | 0.52 | 0.62 | 0.29 |
| detectable_content | 0.85 | 0.89 | 0.79 | 0.89 |
| detectable_format | 0.66 | 0.68 | 0.78 | 0.67 |
| keywords | 0.68 | 0.71 | 0.73 | 0.75 |
| language | 0.68 | 0.58 | 0.87 | 0.97 |
| length_constraints | 0.46 | 0.48 | 0.55 | 0.41 |
| punctuation | 0.24 | 0.14 | 0.17 | 0.11 |
| startend | 0.67 | 0.58 | 0.63 | 0.22 |
| combination | 0.24 | 0.22 | 0.17 | 0.22 |

Table 9: **Success rate** on the IFEvalZhou et al. (2023) across 9 categories of instruction types

| IFEval inst | LLaMa2-chat-7b | LLaMa2-chat-13b | Mistral-7B-inst-v0.3 | Phi-3-mini-128k |
|---|---|---|---|---|
| keywords:existence | 0.79412 | 0.87255 | 0.86275 | 0.94118 |
| keywords:forbidden_words | 0.18627 | 0.28431 | 0.36275 | 0.32353 |
| keywords:frequency | 0.86275 | 0.92157 | 0.91176 | 1.0000 |
| startend:end_checker | 0.23529 | 0.16667 | 0.27451 | 0.13725 |
| detectable_content:number_placeholders | 0.76471 | 0.80392 | 0.5098 | 0.87255 |

Table 10: **Success rate** on IFEval-simple across 5 instruction types under 3 categories

| IFEval inst | LLaMa2-chat-7b | LLaMa2-chat-13b | Mistral-7B-inst-v0.3 | Phi-3-mini-128k |
|---|---|---|---|---|
| change_case | 0.53 | 0.70 | 0.46 | 0.31 |
| detectable_content | 0.65 | 0.90 | 0.75 | 0.94 |
| detectable_format | 0.67 | 0.72 | 0.72 | 0.64 |
| keywords | 0.80 | 0.91 | 0.90 | 0.96 |
| language | 0.40 | 0.10 | 0.94 | 0.83 |
| length_constraints | 0.53 | 0.56 | 0.69 | 0.40 |
| punctuation | 0.15 | 0.25 | 0.06 | 0.00 |
| startend | 0.98 | 0.93 | 0.69 | 0.28 |

Table 11: **Success rate** on IFEval-simple-extended across 8 categories of instruction types (excluding the 'combination' category)

## A.5 PCA ACROSS ALL FIVE INSTRUCTION TYPES

In this section, we extend the PCA analysis to include all five instruction types used in our experiments. This analysis contrasts with the PCA plot in Figure 2 of the main paper, where we focus on three instruction types within the keyword category. In the main paper, the PCA plot show a clear tendency towards separability of the instruction-following dimension across tasks, even though the data points were not perfectly linearly separable. However, in this extended analysis with all five instruction types in Figure 6, the representations are less linearly separable in the 2-dimensional PCA plot. This highlights that different instruction types (or categories) may exhibit distinct geometries in the representation space. The lack of clear separability further supports our findings in the main paper that linear probes trained on one set of instruction types struggle to generalize to unseen instruction types in Section 2.3. This suggests that there is no "global" instruction-following dimension that can be applied across different types of instructions, likely due to the varying internal geometries of these categories.

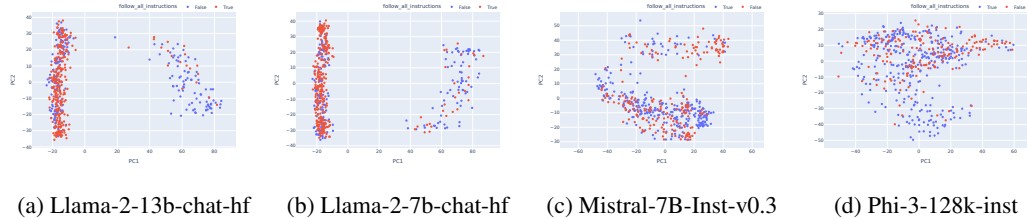

(a) Llama-2-13b-chat-hf   (b) Llama-2-7b-chat-hf   (c) Mistral-7B-Inst-v0.3   (d) Phi-3-128k-inst

Figure 6: **PCA plot of representations from four LLMs across all five instruction types**. This PCA plot of first-token representations from early layers shows that the inclusion of all five instruction types results in less separability compared to the three instruction types in the main paper in Figure 2. This indicates that different instruction types possess distinct geometries, supporting the conclusion that linear probes do not generalize well to unseen instruction types.

## A.6 WHY DO WE CHOOSE IFEVAL DATASET?

Here, we would like to emphasize why we choose IFEval as our primary dataset instead of using real-world dataset with different contexts and domains.

First, we select IFEval to focus on our scope which is 'single, simple, and non-ambiguous instructions'. Real-world datasets often involve complex, ambiguous, or multi-instruction prompts, which

can conflate multiple factors affecting instruction-following. As an initial exploration of the geometry of LLM representations in instruction-following, we chose to focus on single, simple, and verifiable instructions to ensure clarity and disentangle multiple factors. The IFEval dataset is well-suited for this purpose, as it provides 25 distinct types of simple and clear instructions that align with our goal of establishing a robust baseline.

Second, we want to avoid evaluator-induced uncertainties. Most real-world tasks and benchmark datasets rely on LLM-based evaluators to determine whether a response follows an instruction. However, LLM-based evaluators may introduce their own uncertainties or make errors in assessing success or failure, which could obscure our analysis on representations of the tested models. The IFEval dataset avoids this issue by including instructions with deterministic evaluation programs that objectively verify compliance. For instance, an instruction like "please do not include keywords: ..." can be automatically validated using a simple program to check for the presence of those keywords. This feature eliminates ambiguity in evaluation and allows us to isolate the directions related specifically to instruction-following.

One of our main contribution is the careful design of data settings specifically tailored to analyze internal states of LLMs in instruction-following contexts. While IFEval serves as an ideal starting point for this research, we hope our work inspires future efforts to tackle analysis of LLMs in more complex, real-world instruction-following tasks.

## A.7 Reverse Representation Engineering

We conducted initial experiments on reverse representation engineering with two models: Phi-3-mini-128k and Mistral-7B-inst-v0.3. In these tests, we try to move representations towards the failure class by flipping the adjustment vector $-\alpha \times D$

| Model | Original SR | Random SR | Reverse Inst-follow SR |
|---|---|---|---|
| Mistral | $0.58 \pm 0.00$ | $0.56 \pm 0.02$ | $0.54 \pm 0.01$ |
| Phi | $0.71 \pm 0.00$ | $0.63 \pm 0.04$ | $0.60 \pm 0.02$ |

Table 12: Success rates for various models under different settings.

Notably, we set the values conservatively to keep the quality ratio (QR) of reverse RE remains similar to that of random directions (0.86 for Mistral and 0.77 for Phi). The results indicate that the success rate (SR) for reverse RE is worse than random directions, as expected, but the difference is not significant. We anticipate that finding on a validation set will amplify the difference between reverse and random directions. We plan to conduct additional experiments to refine $\alpha$ and better evaluate the effectiveness of reverse RE in disrupting instruction adherence.

