# OpenReview forum: "Do LLMs ``know'' internally when they follow instructions?"
_ICLR.cc/2025/Conference — ICLR 2025 Poster_

### Official Review · Reviewer_jGSV · 2024-11-02

**Soundness:** 3
**Presentation:** 2
**Contribution:** 2
**Rating:** 5
**Confidence:** 3

**Summary:**

The authors explore whether LLMs have an internal representation related to successful instruction following. They train a linear probe on the models' internal representations to predict success/failure cases. This probe works well when tested on held-out tasks. However, when tested on held-out instructions, the probe fails, suggesting that the representation found is not a general "instruction-following" representation.
Potential practical use: Adding this identified direction to a model’s internal representation improves instruction following -- although it is unclear how far this generalizes.

**Strengths:**

The paper demonstrates a practical use case of their findings. The instruction-following direction improves model instruction-following success. This can be a useful case way to steer models to follow instructions better.

Also, the method could help us detect when a LLM will not follow a particular instruction. We can detect this even before the LLMs generate a response — because the method works on the first response token.

This is step towards scientifically understand key processes in how LLMs complete practical tasks

The paper reproduces results across different models. The paper conducts experiments across multiple models (Llama, Mistral, and Phi-3) and shows strong results on all 3 models.

**Weaknesses:**

To strengthen the claim of usefulness, compare your method to other interventions e.g. few-shot prompting. Do we observe a similar increase in success rate by just few-shot prompting? In general, the claim of practical usefulness could be investigated in much more detail (with more baselines and tests of generalization to other datasets).

 You write: “This discovery also suggests explanations for why LLMs sometimes fail to follow clear instructions and why prompt engineering is often effective, even when the content remains largely unchanged.” You then point to properties of the instruction-following dimension (sensitivity to phrasing modification). I don’t see how this helps *explain* why models sometimes fail to follow clear instructions or why prompt engineering helps.

The authors ask the question “Do LLMs know internally when they follow instructions”, but they don’t specifically answer it. I feel that the title and the abstract imply yes. But on closer reading, only on page 4, we find that it does not generalize across instruction types.

My impression is that more recent LLMs (e.g. the latest from OpenAI, Anthropic, Meta) are much more reliable at instruction-following and less sensitive to prompts than the models tested here. This is presumably not due to exploiting the instruction-following direction. More generally, it is unclear whether the insights gained from these weaker models would carry over to more recent models. (I’m not expecting the authors to conduct white-box experiments with much larger open models such as Llama-3-405b but it might be possible to gain evidence on this question in other ways).

Smaller points:
1. “we demonstrate that this dimension generalizes to unseen tasks, indicating that it captures a fundamental aspect of instruction adherence in LLMs.” This claim is ambiguous. What level of generalization performance justified something being a “fundamental aspect”? It would be helpful to make more concrete and falsifiable claims.

2. I find the claim related to Fig 4 questionable. The authors claim that phrasing plays a critical role in accuracy because the phrasing modification direction aligns more with the instruction-following direction. However, they don’t empirically demonstrate that the phrasing changes affect success rates more than the other factors. To make the argument stronger, consider measuring the impact of each factor. How much does the success rate change when modifying phrasing vs difficulty vs task type?

**Questions:**

Define “know internally”. The title and various parts of the paper reference this. After reading, I understand it as “encoded instructions in a way correlated to correct responses”. But perhaps the authors disagree with this definition. Defining it specifically would clarify this.

From above: The authors ask the question “Do LLMs know internally when they follow instructions”, but they don’t specifically answer it. I feel that the title and the abstract imply yes. But on closer reading, only on page 4, we find that it does not generalize across instruction types. So the internal representation seems an encoding of the specific instruction type, rather than a general knowledge of instruction following. Clarifying this earlier would be useful. Readers skimming the paper may have a wrong impression.

---

> ### Author Response · Authors · 2024-11-17
>
> We appreciate your recognition of the value in our study, particularly in uncovering the instruction-following direction as a practical tool for steering models toward better adherence to instructions. Your acknowledgment of its potential to detect instruction-following failures before response generation underscores a key application of our findings. We are also grateful for your recognition of our contribution to advancing the scientific understanding of LLM behavior and for highlighting the robustness and reproducibility of our experiments across multiple models.
>
> ---
> ## W1. Comparing with few-shot prompting
> Thank you for this insightful suggestion. Our study specifically focuses on *simple and non-ambiguous instructions* to isolate the instruction-following dimension under controlled settings. Prompting methods like few-shot prompting or Chain-of-Thought (CoT) are designed to address more complex scenarios, such as ambiguous instructions [1] or tasks requiring multi-step reasoning. Since the instructions in the IFEval dataset are clear and straightforward, we did not include these advanced prompting techniques as baselines.
>
> That said, comparing representation engineering (RE) with few-shot prompting could indeed be an interesting direction for future work. Such experiments might uncover unique benefits or limitations of RE, even within the context of simple, clear instructions. We appreciate your suggestion and will highlight this as a potential avenue for further exploration in our revised manuscript.
>
> [1] Mann, Ben, et al. "Language models are few-shot learners."(2020).
>
> ## W2. Clarification
> > *I don’t see how this helps *explain* why models sometimes fail to follow clear instructions or why prompt engineering helps.*
> >
> Thank you for highlighting this important point. Let us clarify how our findings connect to these observations. Our study uncovered a linearly separable geometry in LLM representations for success and failure cases, with a sensitivity to phrasing modifications. This sensitivity suggests that the LLM’s internal representation space is not robust to small changes in phrasing, similar to how deep neural networks without adversarial training are sensitive to small perturbations on images.
>
> This sensitivity explains two phenomena:
>
> **1)Why LLMs fail to follow clear instructions** Even when instructions are clear and simple, slight differences in phrasing in the input space can shift the internal representations to a failure class in the representation space. This arises from the limitation of LLMs; non-robust geometry of their representation space.
>
> **2)Why prompt engineering helps** Since the instruction-following dimension is sensitive to phrasing, when prompt engineering changes phrasing, as a result, it can effectively move a failure example into the success class in the representation space.
>
> While our work does not fully explain why this sensitivity exists or how it originates during training, it provides a foundation for understanding and addressing this limitation.
>
> ## W3. Clarification
> >*The authors ask the question “Do ...*
> >
> We appreciate your observation and we will clarify this earlier in the revision.
>
> ## W4. Question on Larger models
> > *My impression is that more recent LLMs are much more reliable ...*
> >
> Thank you for raising this point. We agree that larger foundation models (e.g., recent GPT-4, Anthropic models) tend to be more robust to phrasing modifications, likely due to their extensive training on massive and diverse datasets that include a wider variety of instruction forms, improving both robustness and generalizability [1]. Our analysis focuses on models with comparably smaller models, which are widely adopted for on-device applications and fine-tuning. However, we acknowledge that insights from these models may not fully capture the behavior of larger, state-of-the-art models. Extending our analysis to larger models is indeed a promising direction for future work, and we thank the reviewer for emphasizing this important area of inquiry.
>
> [1] Zhu Kaijie, et al. "Promptrobust: Towards evaluating the robustness of large language models on adversarial prompts." (2024).
>
> ## Q2. Define “know internally”
> Yes, your interpretation is correct. By “know internally,” we mean that the model encodes information in its representations that correlates with whether its response will follow the given instruction. This terminology aligns with prior work [1] analyzing hallucination, which refers to similar separability in the representation space for truthful versus hallucinated outputs. We will clarify this definition explicitly in the paper to avoid confusion. Thank you for highlighting this point.
>
> [1] Amos Azaria et al. The internal state of an llm knows when it’s lying. 2023.
>
> ---
>
> We sincerely thank the reviewer for their thoughtful feedback and constructive suggestions. We hope our responses have addressed your questions and clarified the scope and implications of our work.

---

> ### Comment · Reviewer_jGSV · 2024-11-23
> **Response**
>
> W1. I still think few-shot would be a good comparison to run.
>
> W2.
> >While our work does not fully explain why this sensitivity exists or how it originates during training, it provides a foundation for understanding and addressing this limitation.
>
> The first part of this sentence is crucial. I think more work is needed to explain these things.
>
>
> W4.
> I think frontier models like Claude Sonnet 3.5 are much more robust to paraphrasing prompts. This undermines the explanatory potential of this work -- which finds representations that are sensitive to prompt phrasing.
>
> While some of the rebuttals were helpful, they didn't resolve the most important issues raises in the review. Hence I keep the same score.

---

> > ### Author Response · Authors · 2024-11-27
> >
> > Thank you for your thoughtful feedback. We appreciate your valuable insights for future directions.
> >
> > ---
> >
> > ### **W2. Sensitivity in Representations**
> >
> > Regarding sensitivity to phrasing, in this work, we focus on the search of an instruction-following direction in the activation space and its relationship with prompt phrasing in the input space. Our claim based on empirical observations is that, while there exist an instruction-following direction for a given instruction, it is not robust to minor changes in phrasing—akin to the susceptibility of neural networks to adversarial perturbations. We agree that there could be a large number of other important factors that affect instruction following and its sensitivity, including structure in the model weights, distribution of the training data and hyperparameters used during training. However, we believe studying them here would be outside the scope of this work, and we reserve them as future directions. To the best of our knowledge, we are the first work to explore this problem from the perspective of latent space, hoping that this serves as a starting point for further exploration of these questions.
> >
> > ### **W4. Frontier Models and Robustness**
> > Our study focuses on smaller, open-source models commonly used for on-device applications and fine-tuning, which remain critical to understanding instruction-following behavior. Our experiments on robustness to prompt paraphrasing holds true for two different model sizes, 7B and 13B. While it’s unclear whether larger frontier models, such as Claude Sonnet 3.5, exhibit greater robustness to paraphrasing due to extensive training on diverse datasets, we expect our claim to hold based on the empirical evidence. We reserve this analysis to larger frontier models for future work.
> >
> > ### **W3 and Q2. Clarifications**
> > We have addressed the specific clarifications you requested in the revised manuscript. We hope these additions resolve any ambiguities regarding the points raised.

---

### Official Review · Reviewer_UuRY · 2024-11-04

**Soundness:** 2
**Presentation:** 3
**Contribution:** 2
**Rating:** 5
**Confidence:** 4

**Summary:**

The authors expand prior work on finding meaningful directions in the latent space of transformer language models to find a direction associated with whether models follow a given instruction or not.
- For the analysis, they adapt and expand an existing dataset and study four open-source models (7b to 13b parameters).
- Using linear probes, the authors find directions associated with instruction following across the studied tasks, but not generalizing to hold-out instruction types.
- They show that manipulating the latent representation along the found direction can improve instruction following success and maintain instruction tuning quality for already successful cases.
- The authors find that the found latent space direction is affected by prompt sensitivity/phrasing of the instructions (less so by task complexity and perplexity)

**Strengths:**

- The goal of finding a way to manipulate latent representations to improve the instruction following of language models (or detect whether instructions are not being followed) has the potential for significant usability and safety impact while being novel.
- The paper is very well written and the presentation is generally clear.
- The presented methodology is well structured and each section is clearly motivated by the previous results.
- The authors test four open-source models across many different tasks, demonstrating a certain amount of robustness.
- The authors generally do not overclaim their results and are very transparent.

**Weaknesses:**

The authors do not claim that they find a universal direction in these models that fully represents the idea of instruction following. However, to make a significant contribution via usability and understanding of latent representations, this work would benefit from addressing a few weaknesses to enable more robust future work and enhance the usability of the results.

### Weakness 1

The studied tasks are somewhat general but the instructions are fairly simple and not necessarily representative of general language model uses. I wonder how well the results of this work might generalize: It is also unclear how well instructions are being followed in a general use case. To use the example of the paper: You might ask a health bot to be aware of your left knee injury when coming up with training plans, but does adding the found latent space direction have side effects like avoiding leg training altogether or creating asymmetric training plans leading to other issues?  The results in this paper could be significantly strengthened by finding applications that allow for a more nuanced analysis of how instructions are being followed after representation engineering to demonstrate the usefulness of their technique.

### Weakness 2

In Sec. 3, I think an opportunity was missed to strengthen the analysis by demonstrating that the found direction also works in a counterfactual way: Can you use the fitted alpha values and directions for the F2T and T2T tests and only flip the sign of the respective alpha value to check if it also works on (intentional) T2F and F2F? I think this is necessary to demonstrate a minimum of reliable usability and support the claim of safety impact (to potentially get a model to not comply).

### Weakness 3

It is unclear (to me) if using only one direction is sufficient for modeling whether an instruction is being followed or not and even if, whether the presented analysis finds that direction. For example, the prompt sensitivity studies in Sec. 4 affecting the performance could be caused by other meaningful "directions" being entangled in the one you care about. To be more precise: If changing "write a resume for [a] software engineer [...]" to "I want you to write about [a] software engineer resume [...]" makes a difference, how do we know this is not caused by, e.g., confounding the learned directions with the instruction data set with a "dialog in first-person" direction (not unlikely,  given the colorful interpretations modern SAE approaches find).

To link it to another, more concrete use case: Is it possible to circumvent safety training of a language model using the proposed methodology? Although also only necessary and not sufficient, such a more complex example would demonstrate that using only one direction is powerful enough to be useful.

If one direction is not sufficient, it might be worth testing if a higher-order approach (2D, 3D, ...) might be more robust to find that direction. There are plenty and different works on this, but for example, [1] uses PCA on latent representations to find the one relevant direction to model how small backdoored language models process trigger inputs to switch to toxic text.

[1] Lamparth, M. and Reuel, A. Analyzing and editing inner mechanisms of backdoored language models. arxiv:2302.12461, 2023.

### Weakness 4

In Sec. 3, fitting an alpha for each model and instruction type greatly limits usability, especially given that you would need a cleanly separate dataset for all relevant tasks. This is also more difficult for more realistic tasks mentioned in Weakness 1.

### Weakness 5

This is a minor weakness that did not impact the final review score, but I wanted to explain why I rewarded a 3 instead of the maximum score of 4 for the presentation. I would have rewarded a 4 if figures 2 and 4 had used a larger font size for the axis labels, plot legend, and larger marker sizes. Please make these plots more readable for future versions of this work.

### Weakness 6

In Sec. 3, it is not clear whether and how the authors calibrated the GPT-4-based scoring from 0 to 9 to construct the quality ratio. The results in Sec. 3 could therefore be strengthened by stating how the GPT-4-based analysis is being conducted (how exactly is a quality rating of 4 different than a quality rating of 5, in-context examples or not, ...) and how accurate and reliable it is (True positive rate, false positive rate, ...).

### Weakness 7

The stated uncertainties seem very low, especially in Tables 3 and 4 (negligible/0 uncertainties). I think the claims in this work could be strengthened if other methods, e.g., bootstrap resampling, are being used to determine a base uncertainty beyond uncertainties purely determined from the linear probes.

**Questions:**

Generally, please just respond to my weaknesses above. I'm willing to increase the scores (total, contribution, and soundness), especially if you address the crucial ones related the generalizability and usability with new experiments or strong arguments.

Q1: Can you explain where the "middle" token was for the different prompts? Was it the middle token of each sequence and thus changing the position for each prompt?

### Comments (Not considered for the review)

- For Figure 2, I feel LDA (fitted on training data) might be a better fit than PCA to show separability, given that LDA actively tries to separate the known classes.
- In lines 213-214, you state that the instruction type generalization values are close to chance. Given the stated 1-sigma uncertainties in Table 1, the presented values are not statistically significantly different (depending on the definition, 2-5 sigma) from random chance.

---

> ### Author Response · Authors · 2024-11-17
>
> We sincerely thank the reviewer for their thoughtful assessment of our paper and for highlighting several strengths. We are glad that you found our work novel and impactful.
>
> ---
>
> ## W1. Data scope
> Thank you for your thoughtful comments. We kindly refer you to the general response section.
>
> ## W2. Reverse RE
> Thank you for your insightful question about reverse representation engineering. We conducted initial tests on two models, Phi-3-mini-128k and Mistral-7B-inst-v0.3, using reverse RE to move representations toward the failure class ($-\alpha \times D$).
>
> | Model| Original SR| Random SR| Reverse Inst-follow SR|
> |---------|--------------------|------------------|----------------------|
> |Mistral| 0.58 ± 0.00| 0.56 ± 0.02| 0.54 ± 0.01|
> |Phi| 0.71 ± 0.00| 0.63 ± 0.04| 0.60 ± 0.02|
>
> We used conservative $\alpha$ values to maintain a similar quality ratio (QR) for reverse RE and random directions (QR: 0.86 for Mistral, 0.77 for Phi). As expected, reverse RE yields worse success rates (SR) than random directions, but the difference is not yet significant. We believe optimizing $\alpha$ on a validation set will further emphasize the difference between reverse and random directions. Additional experiments are planned to refine $\alpha$ and better assess reverse RE’s impact on instruction adherence.
>
> ## W3. Suggestion on finding multiple directions
> Thank you for raising this insightful point. Considering multiple directions related to instruction-following is indeed a promising avenue for future research. In our current work, we focus on a single direction associated with success in instruction-following as an initial exploration of the geometry of LLM representations in this context. This approach provides a foundational understanding but does not preclude the possibility of additional meaningful directions being present.
>
> Future work could extend this analysis to multiple directions using techniques such as dictionary-based learning to uncover latent factors in an unsupervised manner, or by incorporating human-defined directions, as demonstrated in the paper you suggested. Expanding RE to consider multiple directions could reduce the risk of task degradation or safety concerns. Also, visualizing these directions using methods like sparse autoencoders (SAE) or other interpretability techniques could yield further insights into the nuanced geometry of instruction-following in LLMs. We appreciate your suggestion and will include these points in the discussion section of our revised paper.
>
> ## W4. Question on parameter alpha
> Thank you for your observation. First, we want to clarify that in our current implementation, we require a single $\alpha$ parameter for each model, which is then applied uniformly across all instruction types. This ensures consistency without needing separate parameter fitting for each instruction type.
>
> Our approach uses a basic form of representation engineering (RE) that relies on $\alpha$ to adjust the representations. However, we believe that integrating more advanced RE methods, such as those proposed in [1], which eliminate the need for explicit parameter search, could address this limitation and improve the practical applicability of RE. We discuss this direction as a promising avenue for future work in lines 534–535.
>
> [1] Jinqi Luo, et al. Pace: Parsimonious concept engineering for large language models. arXiv preprint arXiv:2406.04331, 2024.
>
> ## W6. GPT-4 evaluator
> Thank you for pointing this out. As you pointed out, we acknowledge that the interpretation of raw task quality scores from GPT-4 evaluator may be not clear. Thus, rather than directly using the raw task quality scores, we employ a **Quality Ratio (QR)** metric; the proportion of responses scoring above a threshold 7, where this threshold is decided based on the distribution of GPT-4 quality scores.
>
> ## W7. Suggestion
> Thank you for your suggestion regarding uncertainty estimation. We agree that exploring additional methods, such as bootstrap resampling, could provide a more comprehensive understanding of the underlying variability. As mentioned in our future work section (line 533-534), we plan to apply alternative linear probing techniques to enhance AUROC performance and improve uncertainty quantification. This will further strengthen the robustness of our findings in future iterations of this work.
>
> ## Q1. Clarification
> Thank you for pointing this out. The middle token’s representation, $LLM(x_1, x_2 …, x_n, y_1, y_2 …,y_{m//2})$, where ${x_1, x_2, …, x_n}$ are the  n  tokens in the input prompt and ${y_1, y_2, …, y_m}$ are the m tokens in the response. The middle token’s position varies depending on the length of the response for each specific prompt, as it is determined dynamically as m//2 . We will clarify this in the revision.
>
> ---
>
> We sincerely thank the reviewers for their thoughtful feedback and constructive suggestions. We hope our responses have addressed your questions and provided greater clarity on our work.

---

> > ### Comment · Reviewer_UuRY · 2024-11-23
> >
> > Dear Authors,
> >
> > Thank you for your comments and clarifications! Unfortunately, my key concerns are still unresolved. I still doubt the robustness and usability for future work of the conducted study while seeing significant potential.
> >
> > I agree with the authors that the chosen IFEval dataset, consisting of ‘single, simple, and non-ambiguous instructions’, is necessary for initial tests to design a tool to avoid the issues of real-world datasets. However, for the same reason, I doubt that future work can use the results for anything beyond this dataset. So, I believe the authors could significantly strengthen their work by finding at least one application containing some of the complexities of real-world datasets.
> >
> > In addition to the dataset limitation, I still have doubts about the statistical significance of the conducted experiments. For example, the authors only focus on the variance of the linear probe fitting and neglect other sources of uncertainty (how well does the model perform on the base task (e.g., Original SR in table 4), how much noise is added by the GPT-4 evaluator, …). Instead of considering these uncertainties for all experiments, a straightforward approach would be to run a sensitivity study for different sources of uncertainty to demonstrate that the linear probing uncertainty is sufficient to consider as the only source (and then imply that the key results in the initial paper are significant). However, because such experiments are not conducted, I am also questioning the usability of the results.
> >
> > Without further experiments that will directly address these concerns, I am unfortunately not willing to increase my score. I think this work has great potential but is not complete yet.
> >
> > Questions:
> > - You mentioned planned experiments for the reverse RE with a validation set. Can you share the results of these experiments?
> > - Can you clarify how exactly the GPT-4 evaluator threshold is determined from the distribution? The repeated statement in the rebuttal from the paper does not provide sufficient information while also not addressing my concern about how well this approach (cut off at score 7) works as a classifier. Do you have a human-annotated calibration data set (n = 100, which can already be enough) to check if there are false positives or false negatives?

---

> > > ### Author Response · Authors · 2024-11-27
> > >
> > > Thank you for your thoughtful review and constructive suggestions. We greatly appreciate your insights and acknowledge the importance of addressing the concerns you raised regarding data scope and uncertainties. These are indeed valuable directions for future work.
> > >
> > > ---
> > >
> > > ### **Experiments for Reverse RE**
> > > Unfortunately, due to constraints on accessing the developement code, we are unable to run further experiments during the rebuttal period. The results we presented in the rebuttal were obtained during earlier phases of the project. While we included these preliminary results in Appendix A.8 of the updated revision, we plan to conduct further validation experiments and expand this analysis in future iterations of the work.
> > >
> > > ### **GPT-4 Evaluator Threshold and Calibration**
> > > We appreciate your detailed inquiry about the GPT-4 evaluator threshold and its calibration. For this particular evaluation, we use human annotation to determine the cutoff of high-quality versus low-quality samples. Once we obtain 50 samples of GPT-4 quality scores, we compare the the score and the quality via human evaluation, and determine that 7 is the appropriate cutoff for categorizing each sample as either “high-quality” or “low-quality.”

---

### Official Review · Reviewer_wmTS · 2024-11-04

**Soundness:** 3
**Presentation:** 3
**Contribution:** 4
**Rating:** 6
**Confidence:** 3

**Summary:**

This work finds that the representation of token is related to the success rate of instruction following. The authors apply linear probing to identify a specific dimension and use representation engineering (RE) to manipulate the representations. Experimental results show that RE  improves success rate while maintaining quality. Further analysis reveals that phrasing modifications plays a critical role rather than task familiarity or instruction difficulty. This paper provides a deeper understanding to this field.

**Strengths:**

1. This paper provides a new understanding of instruction following, using token representation to interprete LLMs themselves.
2. The experiments are well designed and the writing is easy to follow.

**Weaknesses:**

1. Some of the conclusions in this paper are as expected, e.g. results from Table 1 and Table 2 are related to "Lost in the Middle" phenomenon. [1]
2. The performance improvement brought by this method is limited as shown in Table 4.
3. The IFEval dataset is small and such problem may be addressed by extending instruction following data.

[1] Lost in the Middle: How Language Models Use Long Contexts

**Questions:**

1. Have you tried reverse representation engineering? Will it be worse than random?
2. Why did you focus on the representation of the first token in the last layer? Any insight?

---

> ### Author Response · Authors · 2024-11-17
>
> Thank you very much for your thoughtful review and valuable feedback. We appreciate your recognition of the novel insights provided by our study into instruction-following in LLMs. We are also grateful for your positive feedback on our experimental design and the clarity of our writing.
>
> ---
> ## W1.
> > *Some of the conclusions in this paper are as expected, e.g., results from Table 1 and Table 2 are related to the “Lost in the Middle” phenomenon. [1: Lost in the Middle: How Language Models Use Long Contexts]*
> >
>
> The “Lost in the Middle” phenomenon observed by [1] primarily discusses performance degradation in models due to the position of relevant information in long contexts. In contrast, our prompts are short—typically just two sentences (examples are provided in Tables 6 and 7 in the Appendix). Also, our analysis focuses on token representations during different stages of ‘response generation’ (before, during, and after), rather than different stages of the input prompt (noted in lines 157-158, Section 2.2). Given these differences in focus and context length, we believe our observation in Table 2 addresses distinct aspects of representation use in LLMs. Please let us know if there is an additional aspect of [1] that you believe would be relevant to our work.
>
> ## W2. Question on RE results
> > *The performance improvement brought by this method is limited, as shown in Table 4.*
> >
>
> We want to highlight that our primary aim is to analyze and understand the internal states of LLMs in relation to instruction-following, rather than to introduce a method for improving instruction-following accuracy. In this context, representation engineering (RE) serves as a validation step to assess whether the instruction-following dimension we identified significantly impacts model behavior, as described in Section 3, rather than aiming to  improve instruction-following success accuracy. In our paper, we apply basic form of RE, thus we believe that integrating more advanced RE methods could further improve instruction-following performance and that our findings lay a foundation for this exploration, as we discuss in future work (Section 6.3, lines 534-535).
>
> ## W3. Data scope
> > *The IFEval dataset is small, and this limitation may be addressed by extending instruction-following data.*
> >
>
> Thank you for your thoughtful comments. We kindly refer you to the general response section, where we address the rationale for choosing IFEval as our primary dataset instead of using real-world dataset and the potential to extend analysis to more complex, real-world tasks in future studies.
>
> ## Q1. Reverse RE
> > *Have you tried reverse representation engineering? Will it be worse than random?*
> >
>
> Thank you for your insightful question regarding reverse representation engineering and its impact on instruction-following. We conducted initial experiments on reverse representation engineering with two models: Phi-3-mini-128k and Mistral-7B-inst-v0.3. In these tests, we try to move representations towards the failure class by flipping the adjustment vector ($-\alpha \times D$).
>
> | Model   | Original SR       | Random SR        | Reverse Inst-follow SR |
> |---------|--------------------|------------------|-------------------------|
> | Mistral | 0.58 ± 0.00        | 0.56 ± 0.02      | 0.54 ± 0.01            |
> | Phi     | 0.71 ± 0.00        | 0.63 ± 0.04      | 0.60 ± 0.02            |
>
> Notably, we set the $\alpha$ values conservatively to keep the quality ratio (QR) of reverse RE remains similar to that of random directions (0.86 for Mistral and 0.77 for Phi). The results indicate that the success rate (SR) for reverse RE is worse than random directions, as expected, but the difference is not significant. We anticipate that finding $\alpha$ on a validation set will amplify the difference between reverse and random directions. We plan to conduct additional experiments to refine \alpha and better evaluate the effectiveness of reverse RE in disrupting instruction adherence.
>
> ## Q2. Clarification
> > *Why did you focus on the representation of the first token in the last layer? Any insights?*
> >
>
> Thank you for asking about our methodological choices. We discuss this in Section 3.1 (lines 306–311), and for convenience, we include the explanation below:
> “*we applied adjustments to the representations in the last layer, as it was found to be more robust to variations in $\alpha$. The first token represents the model’s initial input embedding, capturing the model’s state before response generation begins. Since RE aims to adjust internal states early to improve instruction adherence, focusing on this initial representation aligns with our objective.”*  We hope this clarifies our choice.
>
> ---
>
> Thank you once again for your thoughtful feedback and for recognizing the value and clarity of our work. We believe your suggestions will help further enhance the manuscript, and we look forward to incorporating these improvements.

---

### Official Review · Reviewer_mAEy · 2024-11-05

**Soundness:** 4
**Presentation:** 3
**Contribution:** 3
**Rating:** 8
**Confidence:** 3

**Summary:**

This work analyzes and compares LLM internal representations when it can successfully follow instructions, vs. when it fails at following instructions. By training linear probes on representations, authors find a latent direction that predicts instruction following success, and shows it generalizes to unseen tasks and instructions. In addition, this work uses representation engineering to make adjustments along the instruction following dimension they found, and finds these adjustments can improve instruction following performance. Finally, this work also finds that the instruction following dimension is more correlated with how the prompt is phrased, rather than the difficulty of the instruction or the task.

**Strengths:**

This paper is well-structured and the findings are very interesting - while using linear probes to find a latent dimension for a specific phenomenon is not that interesting or original, the follow-up analyses demonstrating that instruction following can be made more accurate with representation engineering, and comparing the correlation of this dimension to different properties of the prompts, are both interesting. In addition, most claims made in the paper are sufficiently backed up with experiments and evidence.

**Weaknesses:**

In section 3, I'm confused what the direction D taken is for different instructions, since the previous section states that linear probes do not generalize well across instructions. Does this mean representation engineering promotes different directions for different instructions, or is there a universal direction that enhances all instructions generally? It would be helpful to clarify this in the paper.
For section 4 as well, one instruction type is chosen to perform the analysis on correlations. Could you elaborate on how much the findings in this section generalizes to other instruction types? Or could there be instruction types whose representation is more correlated with task familiarity rather than phrasing?

**Questions:**

- At the start of section 2, it would be helpful to elaborate what the IFEval dataset is used for
- Make text in Figure 1 bigger

---

> ### Author Response · Authors · 2024-11-17
>
> We really appreciate for your positive and constructive feedback and recognizing the strengths of our work. We are pleased that you found the paper well-structured and appreciated our in-depth follow-up analyses using representation engineering, which go beyond identifying a latent dimension to demonstrate its practical implications. We also value your recognition that our claims are robustly supported by experiments and evidence, underscoring the novelty and significance of our findings.
>
> ---
> ## W1. Clarification on the direction $D$
> > *In section 3, I'm confused what the direction $D$ taken is for different instructions, since the previous section states that linear probes do not generalize well across instructions. Does this mean representation engineering promotes different directions for different instructions, or is there a universal direction that enhances all instructions generally? It would be helpful to clarify this in the paper.*
> >
>
> Thank you for this thoughtful question. In Section 3, we use a universal direction $D$, derived from training on all IFEval-simple data, to enhance instruction-following performance across all instruction types. While $D$ is evaluated on unseen instruction types in Section 2 to test generalizability, in Section 3, it is applied to the same instruction types seen during training, ensuring consistent performance improvements within this scope. We will clarify this distinction in the revised version to ensure greater clarity and address any potential confusion.
>
> ## W2. Clarification on terminology
> > *For section 4 as well, one instruction type is chosen to perform the analysis on correlations. Could you elaborate on how much the findings in this section generalizes to other instruction types? Or could there be instruction types whose representation is more correlated with task familiarity rather than phrasing?*
> >
>
> Thank you for raising this insightful point. Exploring how sensitivity to different interventions, such as phrasing modifications, task familiarity, or instruction difficulty, varies across different instruction types is indeed an interesting direction. We will note this as a promising avenue for future work.
>
> ## Q1 & 2. Suggestion
> > *At the start of section 2, it would be helpful to elaborate what the IFEval dataset is used for and Make text in Figure 1 bigger*
> >
>
> We appreciate this suggestion. We will add a brief explanation of the IFEval dataset’s role at the start of Section 2 to provide better context. Additionally, we will adjust the figure to improve readability.
>
> ---
>
> Once again, thank you for your thoughtful feedback and positive evaluation of our work. We look forward to incorporating your comments to enhance clarity and presentation.

---

> > ### Comment · Reviewer_mAEy · 2024-12-03
> >
> > Thank you for your response, I appreciate it! As my review is already positive, I will maintain my current score.

---

### Official Review · Reviewer_MsZm · 2024-11-06

**Soundness:** 3
**Presentation:** 3
**Contribution:** 4
**Rating:** 6
**Confidence:** 4

**Summary:**

This paper investigates the internal representation of LLMs and identifies a "dimension" that corresponds to their success in instrution-following. They show that manipulating this dimension could improve success in instruction-following without degrading the quality of the outputs. They also measure correlation of the dimension with different perturbations and show that it is more sensitive to the phrasing of the prompt than the task difficiulty or faimiliarity.

'

**Strengths:**

* Paper offers very interesting and operationalizable insights about the internal representations of LLMs.
* Paper tells a very good, satisfying story: from identifying the dimension, to manipulating and interpreting it.

**Weaknesses:**

- The methodology description lacks clarity due to insufficient mathematical notation. It is especially difficult to follow the procedures outlined in Sections 3 and 4. The authors should clearly define mathematical objects, such as "input representation," and provide precise equations for the computations (e.g., lines 370-374). Specifically, given a prompt \( x_1, x_2, ..., x_n \) of n words and an output \( y_1, y_2, ..., y_m \) of  m words, what is the "input representation" as a function of these tokens? It appears to be computed solely from the output words.

* The terminology is confusing. The term "instruction" would be more accurately described as "constraints." It’s challenging to distinguish "instruction" from "task," as "instruction" could imply the entire command given to the model. Additionally, the meaning of "dimension" is unclear. Does it refer to a hypothetical concept rather than an actual dimension of a hidden vector computed by the model?

* The experiments were conducted on a single, relatively small dataset, raising questions about whether the findings will generalize.

**Questions:**

- See weaknesses
- What is the motivation behind the update in line 303-304? Could you provide more intuition on what is being done to the input representation there?

---

> ### Author Response · Authors · 2024-11-17
>
> Thank you very much for your thoughtful review and valuable feedback. We appreciate your constructive comments, highlighting that our paper provides valuable, actionable insights into the internal representations of LLMs and effectively builds a coherent narrative, from identifying the instruction-following dimension to manipulating and interpreting it.
>
> ---
> ## W1. Notation
> > *The methodology description lacks clarity due to insufficient mathematical notation. It is especially difficult to follow the procedures outlined in Sections 3 and 4. The authors should clearly define mathematical objects, such as "input representation," and provide precise equations for the computations (e.g., lines 370-374). Specifically, given a prompt ( x_1, x_2, ..., x_n ) of n words and an output ( y_1, y_2, ..., y_m ) of m words, what is the "input representation" as a function of these tokens? It appears to be computed solely from the output words.*
> >
>
> Thank you for pointing this out. To clarify, the “input representation” refers to the model’s representation after processing only the input prompt, denoted by $LLM(x_1, x_2 …, x_n)$.
> Here is more details about three representations we analyze in lines 154-161: (1) the first token’s representation, which is before generating the response $LLM(x_1, x_2 …, x_n)$; (2) the middle token’s representation, $LLM(x_1, x_2 …, x_n, y_1, y_2 …,y_{m//2})$, representing the model state halfway through response generation; and (3) the last token’s representation, $LLM(x_1, x_2, …, x_n, y_1, y_2 …,y_m)$, representing the model state after completing the response.
>
> In Section 4 (lines 370-374), we conduct a sensitivity analysis to understand the impact of various prompt modifications on these internal representations. This analysis evaluates how different perturbations (phrasing, task familiarity, instruction difficulty) affect the LLM’s instruction-following capability, described in lines 375-418. We will clarify these definitions and notations in the revision to make these concepts more accessible.
>
> ## W2. Terminology
> > *The terminology is confusing. The term "instruction" would be more accurately described as "constraints." It’s challenging to distinguish "instruction" from "task," as "instruction" could imply the entire command given to the model. Additionally, the meaning of "dimension" is unclear. Does it refer to a hypothetical concept rather than an actual dimension of a hidden vector computed by the model?*
> >
>
> We use the term *“instruction”* to align with the IFEval dataset terminology, referring to specific guidelines or constraints (e.g., “please do not use keywords”), while *“task”* refers to the broader context or content (e.g., “please write a resume”) within which the instruction is applied (lines 51–78). However, we understand that “instruction” could imply the entire command given to the model, which might lead to confusion. To address this, we will revise the text to clarify that the term *“instruction”* specifically refers to constraints applied within a task context.
>
> The *“instruction-following dimension”* refers to an actual direction in the model’s representation space, identified through the weights of a linear probe trained to distinguish between successful and unsuccessful instruction-following instances. To make this clearer, we will adjust the terminology and provide explicit definitions in the revision. Thank you for pointing this out.
>
> ## W3. Data Scope
> > *The experiments were conducted on a single, relatively small dataset, raising questions about whether the findings generalize.*
> >
>
> Thank you for your thoughtful comments. We kindly refer you to the general response section, where we address the rationale for choosing IFEval as our primary dataset instead of using real-world dataset and the potential to extend analysis to more complex, real-world tasks in future studies.
>
> ## Q. Clarification on motivation
> > *Could you explain the motivation and intuition behind the update in lines 303-304?*
> >
>
> Thank you for the question. The update in lines 303-304 involves applying representation engineering to shift each input representation towards the instruction-following direction $D$. Here, $D$ is a vector that points toward successful instruction-following cases in the model’s representation space, while -D points towards failures. By adding $\alpha \times D$ to the original representation $R_{original}$, we effectively shift $R_{original}$ closer to the success class, improving the model’s adherence to instructions. This update leverages the instruction-following direction identified through linear probes, providing a mechanism for improving performance in instruction-following tasks.
>
> ---
>
> We sincerely thank the reviewers for their thoughtful feedback and constructive suggestions. Your insights have helped us refine the clarity and presentation of our work. We hope our responses address your concerns.

---

> > ### Comment · Reviewer_MsZm · 2024-11-19
> > **Thank you**
> >
> > Thanks for the thorough reply. It addresses most of my concerns. The only one that remains is the limitation of the evaluation. Confirming the findings on some more datasets would greatly strengthen the paper, since a single dataset may have artifacts. I decided to maintain my current score.

---

> > > ### Author Response · Authors · 2024-11-27
> > >
> > > Thank you for your thorough review and valuable feedback. We appreciate your acknowledgment of the aspects we addressed. Thank you again for your time and constructive input.

---

### Author Response · Authors · 2024-11-17

# General response
We are grateful for the reviewers’ thoughtful and constructive feedback on our submission. The reviewers highlighted key strengths of our paper, including its novel contributions to understanding instruction-following in LLMs by analyzing their internal representations. This allows us to detect potential instruction-following failures early, even before a response is generated, using early-stage representations. The paper’s well-structured methodology, clear presentation, and rigorous experiments across multiple models were also praised. Reviewers noted that the work tells a “good, satisfying story,” providing a meaningful step toward scientifically understanding how LLMs follow instructions.

---
## Q. Data Scope
> *Reviewer MsZm: The experiments were conducted on a single, relatively small dataset, raising questions about whether the findings will generalize.*
>

> *Reviewer wmTS: The IFEval dataset is small and such problem may be addressed by extending instruction following data.*
>

> *Reviewer UuRY: The studied tasks are somewhat general but the instructions are fairly simple and not necessarily representative of general language model uses. I wonder how well the results of this work might generalize: It is also unclear how well instructions are being followed in a general use case. To use the example of the paper: You might ask a health bot to be aware of your left knee injury when coming up with training plans, but does adding the found latent space direction have side effects like avoiding leg training altogether or creating asymmetric training plans leading to other issues? The results in this paper could be significantly strengthened by finding applications that allow for a more nuanced analysis of how instructions are being followed after representation engineering to demonstrate the usefulness of their technique.*
>

Thank you for your concern about datasets. We do agree it would be beneficial to include real-world datasets to validate the findings. However, here we would like to emphasize why we choose IFEval as our primary dataset instead of using real-world dataset with different contexts and domains.

**First, we select IFEval to focus on our scope which is ‘single, simple, and non-ambiguous instructions’.** Real-world datasets often involve complex, ambiguous, or multi-instruction prompts, which can conflate multiple factors affecting instruction-following. As an initial exploration of the geometry of LLM representations in instruction-following, we chose to focus on single, simple, and verifiable instructions  to ensure clarity and disentangle multiple factors. The IFEval dataset is well-suited for this purpose, as it provides 25 distinct types of simple and clear instructions that align with our goal of establishing a robust baseline.

**Second, we want to avoid evaluator-induced uncertainties.** Most real-world tasks and benchmark datasets rely on LLM-based evaluators to determine whether a response follows an instruction. However, LLM-based evaluators may introduce their own uncertainties or make errors in assessing success or failure, which could obscure our analysis on representations of the tested models. The IFEval dataset avoids this issue by including instructions with **deterministic evaluation programs** that objectively verify compliance. For instance, an instruction like “please do not include keywords: …” can be automatically validated using a simple program to check for the presence of those keywords. This feature eliminates ambiguity in evaluation and allows us to isolate the directions related specifically to instruction-following.

One of our main contribution is the careful design of data settings specifically tailored to analyze internal states of LLMs in instruction-following contexts. **While IFEval serves as an ideal starting point for this research, we hope our work inspires future efforts to tackle analysis of LLMs in more complex, real-world instruction-following tasks (lines 530-532).**

---

### Author Response · Authors · 2024-11-27

# **About the revision**
We sincerely thank all reviewers for their thoughtful feedback and constructive suggestions, which have helped us improve the clarity and scope of our paper. Below, we summarize the key updates we have made in this revision based on the feedback received.

---

### **Clarifications**

**Terminology (Abstract and Instruction)**

We have revised the abstract and introduction to clarify the interpretation of the title *“Do LLMs know internally when they follow instructions”*. Specifically, we define *“know internally”* as the encoding of information in LLM representations that correlates with instruction-following success. Additionally, we clarify that while the observed instruction-following direction generalizes to unseen tasks, it does not generalize to unseen instruction types (Reviewer jGSV).

**Mathematical Notation in Section 2.2**

To improve clarity, we have updated Section 2.2 with precise mathematical notations describing the representations analyzed at different stages of response generation, addressing Reviewer MsZm’s concern about insufficient clarity.

### **Discussion on reviews**

**Discussion on Dataset Scope**

We have incorporated a discussion on the dataset scope from the rebuttal into Section 2.1 and Appendix A.7. This addresses concerns about the generalizability of our findings and explains the rationale for using the IFEval dataset as a controlled benchmark for analyzing instruction-following.

**Reverse Representation Engineering Results**

We have added preliminary results for reverse representation engineering to Appendix A.8. These experiments explore whether reverse adjustments to the instruction-following dimension degrade instruction adherence and provide insights into the robustness of our findings. We plan to extend this analysis in future revisions.

---

We hope these revisions address key points raised by reviewers and enhance the overall clarity and impact of our work. Thank you for your time and consideration.

---

### Meta-Review · Area_Chair_eKQK · 2024-12-19

**Metareview:**

The authors present a way to understand whether instructions will have the desired effect on instruction following behavior. There were concerns about how reliable these results are (relatively small effect) and applicability to modern models that are more robust. This is very much on the borderline, since the effects are not that impressive -- but given the transparency from the authors without over-claiming, and the overall novelty / relevance of the research direction, it could make a good addition to the conference.

**Additional Comments On Reviewer Discussion:**

Significant clarifications and some new experiments were carried out.

---

### Decision · Program_Chairs · 2025-01-22

Accept (Poster)